



# A sprinkling experiment to quantify celerity-velocity differences at the hillslope scale

Willem J. van Verseveld[1], Holly R. Barnard[2], Chris B. Graham[3], Jeffrey J. McDonnell[4, 5], J. Renée Brooks[6], Markus Weiler[7]

[1]Deltares-Catchment and Urban Hydrology Department, Delft, The Netherlands
[2]Institute of Arctic and Alpine Research, Department of Geography, University of Colorado, Boulder, CO, USA
[3]Hetchy Hetchy Water and Power, Moccasin, CA, USA
[4]Global Institute for Water Security and School of Environment and Sustainability, University of Saskatchewan, Saskatchewan, Canada
[5]School of Geoscience, University of Aberdeen, Aberdeen, Scotland
[6]Western Ecology Division, U.S. EPA/NHEERL., Corvallis, Oregon, USA
[7]Chair of Hydrology, Faculty of Environment and Natural Resources, University of Freiburg, Germany

*Correspondence to*: Willem J. van Verseveld (willem.vanverseveld@deltares.nl)

**Abstract.** The difference between celerity and velocity of hillslope water flow is poorly understood. We assessed these differences by combining a 24-day hillslope sprinkling experiment with a spatially explicit hydrologic model analysis. We focused our work at Watershed 10 at the H.J. Andrews Experimental Forest in western Oregon. δ2H label was applied at the start of the sprinkler experiment. Maximum event water (δ2H labeled water) contribution was 26% of lateral subsurface flow at 20 h. Celerities estimated from wetting front arrival times were generally much faster (on the order of $10 - 377$ mm h-1) than average vertical velocities of δ2H (on the order of $6 - 17$ mm h-1). In the model analysis, this was consistent with an identifiable effective porosity (fraction of total porosity available for mass transfer) parameter, indicating that subsurface mixing was controlled by an immobile soil fraction, resulting in an attenuated δ2H in lateral subsurface flow. Furthermore, exfiltrating bedrock groundwater that mixed with lateral subsurface flow captured at the experimental hillslope trench caused further reduction in the δ2H input signal. Our results suggest that soil depth variability played a significant role in the velocity-celerity responses. Deeper upslope soils damped the δ2H input signal and played an important role in the generation of the δ2H breakthrough curve. A shallow soil (~ 0.30 m depth) near the trench controlled the δ2H peak in lateral subsurface flow response. Simulated exit time and residence time distributions with the hillslope hydrologic model were consistent with our empirical analysis and provided additional insights into hydraulic behavior of the hillslope. In particular, it showed that water captured at the trench was not representative for the hydrological and mass transport behavior of the entire hillslope domain that generated total lateral subsurface flow, because of different exit time distributions for lateral subsurface flow captured at the trench and total lateral subsurface flow.





## 1 Introduction

Residence time distributions, velocities and celerities at the catchment and hillslope scale are poorly understood (McDonnell and Beven, 2014). Countless studies have shown that the hydrologic response of headwater catchments can be much faster (orders of magnitude) than the mean transit time of water (e.g. Martinec, 1975; Seeger and Weiler, 2014). Or, in other words

during a rainfall or snowmelt event stream discharge responds immediately, while the average age of stream water itself is can be years or decades (Kirchner, 2003). Since different mechanisms control celerities, or the hydrograph response to perturbations, and flow velocities, or the time it takes a tracer to travel through the system, differences between the two are to be expected (McDonnell and Beven, 2014). Nevertheless, few studies have quantified these differences and have explained the processes responsible for it.

Although many studies did find different controls on how stored water rapidly travels to the stream; transmissivity feedback (Kendall et al., 1999; Bishop, 1991), via macropore flow (McDonnell, 1990), pressure wave propagation (Torres et al., 1998; Williams et al., 2002) and hydrodynamic mixing (Jones et al., 2006), open questions remain in quantifying the difference between celerity and velocity. This is because it has been difficult to add tracer to a sufficiently large area of a watershed to force the differentiation of celerity and velocity in measured flow and tracer response. For instance, Bishop et

al. (2004) studied old water contribution to streamflow during storm events, and because of these natural conditions, the natural variability in isotope inputs added uncertainty to their results. It is difficult to characterize under rainfall conditions the natural isotope input due to spatial and temporal heterogeneity in rainfall and snowmelt inputs at the catchment scale. Frequently, the isotope input is only sampled at one location in the study area which can have considerable effect on the uncertainty in modeled residence time and in hydrograph separation (McGuire and McDonnell, 2006). While the study of

Anderson et al. (1997) did not have issues with natural variability in inputs because of controlled sprinkler conditions, during their sprinkler experiment only shallow soil lysimeters showed deuterium tracer peaks and soil water breakthrough curves were incomplete. With incomplete breakthrough curves it is a challenge to determine transport process and associated flow velocities, as has been examined in several plot to hillslope scale studies (e.g. Feyen et al., 1999; Weiler et al., 1998; Wienhofer et al., 2009).

To date, the focus of many experimental studies concerning the tracer transport at the hillslope scale has been the qualitative and/or conceptual descriptions of velocities, observed transit times and dispersion of solutes (e.g. Anderson et al., 1997; Nyberg et al., 2003; Jackson et al., 2016). Several studies interpreting the transport of solutes in a more quantitative way have relied on a lumped modeling approach (e.g. Rodhe et al., 1996; Jury and Sposito, 1985), assuming time invariant flow conditions. Application of flow weighted rescaling techniques in the case of a variable flow system relies on assumptions

that are rarely met in the field (Rodhe at al., 1996; Rinaldo et al., 2011). More recently, and increasingly, coupled hydrological and solute transport models have been used in a spatial explicit manner (e.g. Vache and McDonnell, 2006; Dunn et al., 2007; McGuire et al., 2007; Sayama and McDonnell, 2009; Davies et al., 2013; van der Velde et al., 2010, 2012) or by using a lumped approach (e.g. Fenicia et al., 2010; Botter et al., 2010; Hrachowitz et al., 2013; Klaus et al., 2015) to



examine the solute (or tracer) response in the stream or in lateral subsurface flow. Sayama and McDonnell (2009) time-space accounting scheme (T-SAS) modeling results suggested that a catchment with thicker soil depth reveals a more damped and longer stream mean residence time (MRT) in response to rainfall. Van der Velde et al. (2010) showed with their solute transport model the importance of including time-varying travel time distributions to understand observed nitrate

concentration dynamics. Dunn et al. (2007) used a series of virtual experiments to examine primary control (hydrological processes and geographical) on the stream water mean residence. Davies et al. (2013) recently developed the multiple interacting pathways (MIPs) model, a particle based model (particle tracking), that includes different velocity pathways and exchanges between these pathways, for exploring flow and transport processes. Yet, realistic and consistent representation of both hydrograph shapes and solute transport remains a grand challenge in hydrology. New datasets that combine tracer data

and hydrograph information are needed to improve our understanding of controls on transit times and transit time distributions (McDonnell and Beven, 2014; Klaus et al., 2015).

Here, we mechanistically assess the differences between the celerity and velocity of the hillslope hydrograph using a combination of a hillslope-scale sprinkler experiment and a process-based spatially explicit hydrologic model. We conducted a 24-day sprinkler experiment where we added a conservative tracer deuterium ($\delta^2$H) at the start of application. This

approach avoided many of the variability issues that occur during natural rainfall by carefully monitoring input chemistry and application rates. Steady sprinkler rates and the long sprinkler experiment duration allowed us to examine complete breakthrough curves of $\delta^2$H in and lateral subsurface flow, groundwater and soil water, and enabled us to experimentally construct transit time distributions. Using this approach, we addressed the following questions:

     1. What is the shape of the hillslope hydrograph and tracer breakthrough?

2. What are the velocities, celerities and transit time distributions of the hillslope hydrograph?

     3. How do our findings compare to recent theoretical work addressing transit time distributions (e.g. Botter et al., 2010; Rinaldo et al., 2011; Harman, 2015)?

## 2 Site description

This study was carried out in Watershed 10 (WS10). WS10, located on the western boundary of the H. J. Andrews

Experimental Forest (HJA), in the west-central Cascade Mountains of Oregon, USA (44.2° N, 122.25° W) ,is a headwater catchment and has an area of 10.2 ha (Figure 1). The climate of HJA is Mediterranean, characterized by dry summers and wet mild winters. Mean annual precipitation is 2220 mm. Eighty percent of precipitation falls between October and April during long duration storm events characterized by low rainfall intensities. While in WD10 snow accumulations are not uncommon, they rarely exceed 0.3 m, and usually melt within 2 weeks (Sollins et al., 1981). At the watershed outlet the

elevation is 470 m and increases at the southeastern ridge line to 680 m. Harvesting took place in WS10 during May-June 1975 and naturally regenerated second-growth Douglas-fir (*Pseudotsuga menziesii*) forest now dominates. Along the stream several seep areas have been identified by Harr (1977) and Triska et al. (1984). The location of these seep areas is controlled



by the presence of vertical, approximately 5 meters wide andesitic dikes, within the southern aspect of the hillslope, and the local topography of bedrock and/ or saprolite (Swanson and James, 1975; Harr, 1977).

Our study hillslope is situated 91 m upstream from the WS10 stream gauging station, on the south aspect of the watershed (Figure 1). The stream-to-ridge slope is 125 m long and is characterized by an average gradient of 37º.The gradient ranges from 27º near the ridge to 48º adjacent to the stream (McGuire, 2004). The elevation range of this site is 480 to 565 m. The volcanic bedrock includes andesitic and dacitic tuff and coarse breccia (Swanson and James, 1975). At the stream-hillslope interface, the depth to unweathered bedrock ranges from 0.1 to 0.6 m and the depth increases gradually toward the ridge to a depth of approximately 3 to 8 m. The soils formed either in residual parent material or in colluvium originating from these deposits and are about 1 m deep. Soil textures show little variation with depth. The surface soils are mostly identified as gravelly loams, lower soil layers are characterized by gravelly silty clay loams or clay loams and subsoils are characterized by gravelly loams or clay loams (Harr, 1977). The highly andic soils vary across the landscape as either Andic Dystrudepts or as Typic Hapludands (Yano et al., 2005). Below the soils the relatively low permeability subsoil (saprolite) of 1-8 m formed from the highly weathered coarse breccia (Ranken, 1974; Sollins, 1981). The well aggregated surface soils show, at lower depths (70-110 cm) a more massive blocky structure with a lesser amount of aggregation than surface soils (Harr, 1977).

## 3 Methods

### 3.1 Sprinkler experiment and instrumentation

To measure lateral subsurface flow, a 10 m long trench was constructed (McGuire et al., 2007) at a natural seepage face. A calibrated 30º V-notch weir collected routed intercepted subsurface water from the trench. Stage at the 30º V-notch weir was recorded at 10 min time intervals by a 1 mm resolution capacitance water-level recorder (TruTrack, Inc., model WT-HR). The duration of the sprinkling experiment was 24 days. The sprinkler experiment started on Day of Year (DOY) 208 (July 27), 2005, directly upslope from the trench on an approximately 8 m by 20 m section of the study hillslope (Figure 1). The sprinkler system consisted of 36 micro-sprinklers spaced in a 2 m grid. An automatic timer sustained a consistent input rate of irrigation water throughout the sprinkler experiment. Irrigation water was stored in forest fire tanks at a landing near the ridge, 300 m from the hillslope. This water was transported to the hillslope by fire hose with two pressure regulators that provided constant head. Irrigation rates were recorded with three tipping bucket rain gauges (TruTrack, Inc., model Rain-SYS-1mm) throughout the experiment. Irrigation rates were also measured during DOY 219 through 226 of the experiment by 72 (0.05 and 0.10 m diameter) randomly placed cups. The cups were sampled every 4-12 h. We applied water with an average rate of 3.7 mm h$^{-1}$ between days 208 and 232 continuously except for 9 h on day 210 with no application of water, and on day 226, 228, 229 and 230 with higher irrigation rates due to timer malfunctions. We applied a pulse of deuterium-enriched water for 24.5 h, starting on DOY 208, to the study area via the sprinkling. A 20,000 L reservoir with a natural





isotopic signature ($\delta^2$H = -69 ‰) was mixed with approximately one liter of 99.8 % of $D_2O$ and provided a source of deuterium with a $\delta^2$H signature of 285

Three groups of porous cup suction lysimeters (Soil Moisture Equipment Corp., Model 1900, 2 bar) were installed at 0.3, 0.7 and 1.1 m depths at site BL and DL. The deepest lysimeter at site AL was installed at 0.8 m depth (soil bedrock interface)

(Figure 1). We used three zero tension lysimeters of 0.15 m x 0.15 m. One zero tension lysimeter collected water at 0.2 m depth, and two zero tension lysimeters collected water from the organic horizon. In an upslope transect twenty-five superquartz (Prenart Equipment ApS) 50 kPa tension lysimeters were installed at a 30° angle (Lajtha et al., 1999), at shallow (0.2 m), middle (0.3-0.4 m), and deep (0.7-0.8 m) soil depths. We measured soil matric potential with seven fast responding tensiometers (type: UMS T4, 1 bar porous cups). The tensiometers were installed vertically in an upslope transect with two

clusters of tensiometers at 0.3 and 0.7 m depth, and one cluster of tensiometers at 0.3, 0.7 and 1 m depth.

As part of an earlier study at this site (van Verseveld et al., 2008), 69 maximum rise wells (0.032 m diameter) were installed to measure water table at the soil-saprolite or bedrock interface. Wells were augered with a hand auger until refusal. The wells were screened for the lower 0.25 m, which was the maximum water height observed by Harr (1977). Of these 69 maximum rise wells, 31 wells were located in the sprinkler plot. Two wells A05 and E04 that were installed at depths of 0.45

and 1.25 m respectively, and located in the sprinkler plot showed transient water table (Figure 1) during a previous study at this site (van Verseveld et al., 2008). One well A01 that was installed at a depth of 0.35 m, located in a groundwater seep (Figure 1) showed deep groundwater during a previous study at this site (van Verseveld et al., 2008). Before the installation of wells a cone penetrometer survey was conducted, that showed that well A01 was the only well located at the "soil-bedrock interface", while the depth to bedrock at well A05 and well E04 was at least 0.8 and 2m, respectively. All

instrument locations are shown in Figure 1.

We measured soil volumetric water content within the irrigated area with 24 time domain reflectometry (TDR) sensors (Environmental Sensors, Inc., model PRB-A). Each TDR sensor was 1.2 m in length and measured at the following five depth intervals: 0-0.15 m, 0.15-0.3 m, 0.3-0.6 m, 0.6-0.9 m, and 0.9-1.2 m. The TDR sensors were installed in a 4 (parallel to the stream channel) by 8 grid (perpendicular to the stream channel), with a spacing of 2 m in each direction. Measurements

were recorded every hour for the duration of the experiment. Fifty-two TDR sensor segments of the total of 120 segments (locations x depth) gave consistent results. Inconsistent results were caused by sensor segments that were not completely installed in the soil profile, reduced contact between the sensor and soil or poor electrical connection. We only included data from sensors that gave consistent results in our analysis. Besides TDR sensors, soil water content was measured with water content reflectometers (WCR) (CS615, Campbell Scientific, Inc.). We installed these soil moisture probes at three different

depths of 0.3 m, 0.7 m and 1.0 m, parallel to the slope, in three soil pits located 15, 20 and 25 m from the slope base (McGuire and McDonnell, 2010). The WCR probes were calibrated using soil cores that were obtained from a number of locations at the H. J. Andrews, including WS10 (G. W. Moore et al., unpublished data, 2003; Czarnomski et al., 2005).



### 3.2 Sampling and chemical analysis

Lateral subsurface flow at the hillslope trench was sampled every 2 to 4 h with an automated ISCO sampler. From DOY 208 to 218, zero tension and tension lysimeters were sampled daily, while from DOY 219 until DOY 235, lysimeters were sampled every other day. We did evacuate the tension lysimeters to -50 kPa and allowed the tension lysimeters to collect

water for 24 h. Transient water table was sampled daily from three wells (A05, E04 and A01) (Figure 1).

The analysis of water samples for hydrogen isotope ratios was performed on (1) an isotope ratio mass spectrometer (Delta plus, ThermoQuest Finnigan, Bremen Germany) interfaced with a high temperature conversion/elemental analyzer (TC/EA, ThermoQuest Finnigan) and (2) a liquid water isotope analyzer using off-axis integrated cavity output spectroscopy (Los Gatos Research, Inc). All hydrogen isotope ratios are expressed as $\delta^2$H values relative to Vienna-standard mean ocean water

(V-SMOW) in ‰:

$$\delta^2 H = \left( \frac{R_{sample}}{R_{standard}} - 1 \right) \qquad (1)$$

where $R_{sample}$ is the ratio of deuterium to hydrogen atoms of the sample and $R_{standard}$ is the standard (V-SMOW). Measurement precision was 1.1‰ for the isotope ratio mass spectrometer and 0.4‰ for the liquid water isotope analyzer.

### 3.3 Two component mixing model

A two-component mixing model was applied to the $\delta^2$H breakthrough in lateral subsurface flow:

$$Q_t C_t = Q_p C_p + Q_e C_e, \qquad (2)$$

where $Q$ is lateral subsurface flow from the trench, $C$ is the measured $\delta^2$H, and the subscripts $t$, $p$ and $e$ stand for total lateral subsurface flow, pre-event water and event water, respectively. Pre-event water was defined as subsurface flow before the start of adding deuterium enriched water to the sprinkler plot.. Unlabeled sprinkler water was also defined as pre-event

water. Because unlabeled sprinkler water and subsurface flow before the start of adding deuterium enriched water had the same $\delta^2$H signal (-75 ‰), a two-component mixing model was sufficient. Event water was defined as the average $\delta^2$H of deuterium enriched water ($C_e$).

### 3.4 Velocities and celerities

The average vertical velocity of labeled water was calculated by assuming that the observed $\delta^2$H peak in wells and lysimeters

coincided with the passage of the $\delta^2$H peak in sprinkled water (consistent with the method of Anderson et al. (1997)).

$$v = \frac{d}{\Delta t} \qquad (3)$$





where $d$ is depth of lysimeter or well, and $\Delta t$ is the travel time between the time of the center of mass of $\delta^2H$ in sprinkler water and $\delta^2H$ peak in soil- or groundwater.

Wetting front celerities were estimated from TDR, groundwater level and tensiometer measurements. The arrival time of the wetting front was defined as the first significant response at the depth of measurement to irrigation. The depth of

measurement divided by the corresponding arrival time provided the wetting front celerities.

### 3.5 Hillvi model

*Hillvi* is a spatially explicit model for water flow and solute transport at the hillslope scale with an explicit coupling between the unsaturated and saturated zone (Weiler and McDonnell, 2004, 2005, 2007; McGuire et al., 2007; Tromp-van Meerveld and Weiler, 2008). The unsaturated and saturated zone coupling has been implemented in the model because several field

studies have shown the importance of the conversion of the unsaturated zone water to transient saturation during storm events (Harr, 1977; McDonnell, 1990; Tani, 1997). We provide a brief description of the model and detailed descriptions can be found in Weiler and McDonnell (2004, 2005, 2007) and an application in the same watershed in McGuire et al. (2007). The unsaturated zone is characterized by a time variable water content between the soil surface and the time variable water table. . The size of the saturated zone is defined by the elevation difference between the water table and an impermeable or

semi-impermeable bedrock surface and the porosity. Lateral subsurface flow within the saturated zone is calculated using the Dupuit-Forchheimer assumption. Itis routed downslope using the approach of Wigmosta and Lettenmaier (1999), according to the local water table gradient between adjacent grid cells. *Hillvi* uses an exponential depth function for drainable porosity and saturated hydraulic conductivity, a parameter that controls transient water development (Weiler and McDonnell, 2004). An exponential decline of drainable porosity and saturated hydraulic conductivity with soil depth is typical for Watershed

10, based on the hydrological properties analysis of 452 soil cores by Harr and Ranken (1972), Ranken (1974) and Harr (1977). The unsaturated zone water balance is controlled by the rainfall that enters the unsaturated zone, losses as vertical recharge and bypass flow (optional) and actual evapotranspiration from the unsaturated zone, and change in storage (water content). The amount of bypass flow is a function of precipitation rate and soil moisture (McGuire and McDonnell, 2007). Actual evapotranspiration is estimated by the potential evapotranspiration and the relative water content in the unsaturated

zone. The water balance of the saturated zone is controlled by inputs as recharge and bypass flow (optional) from the unsaturated zone and the lateral inflow and losses as outflow, lateral pipeflow (optional), seepage to bedrock (optional) and the change in storage (water table depth). Seepage to bedrock is a function of the saturated hydraulic conductivity of the bedrock and the total head above the bedrock (Tromp-van Meerveld and Weiler, 2008).

Solute flux in recharge and bypass flow is determined by the recharge and bypass flow of a grid cell and the average

concentration in the unsaturated zone. An effective porosity coefficient (fraction of total porosity) is used to describe the available pore volume for mass transfer. This effective porosity coefficient is used to account for the potential influence of immobile water on tracer movement. Lateral subsurface solute fluxes are determined by taking the average concentration in the saturated zone available for solute transport and multiply it with the subsurface flow. . The transfer of solutes between



the unsaturated zone and saturated zone depends on whether the water table is rising or falling and the total porosity minus the drainable porosity. The amount of solutes that are transported from the saturated to the unsaturated zone, under a falling water table depends on the concentration in the saturated zone, how much the water table falls and the total porosity minus drainable porosity (Weiler and McDonnell, 2004). The amount of solutes that are transported from the unsaturated to the

saturated zone, under a rising water table depends on the average concentration in the unsaturated zone (Weiler and McDonnell, 2004). The mass flux of seepage to bedrock is controlled by the average concentrations in the saturated zone and seepage rate to bedrock. Complete mixing in each grid cell and each zone is assumed when calculating the saturated and unsaturated zone concentrations (Weiler and McDonnell, 2004).

The model of study hillslope was setup using a DEM (1x1 m) based on the surface topography and soil depth survey. Inputs

to the model were irrigation rates, canopy reference evapotranspiration (CRET) and deuterium concentration of irrigation water. CRET was estimated with the standard Penman–Monteith equation for canopy reference evapotranspiration using meteorological measurements from a weather station placed in the irrigated area. We refer to Barnard et al. (2010) for detailed information about the CRET calculation. A simple mixing model was applied to the simulated deuterium concentrations in lateral subsurface flow from the trench with *Hillvi* to allow for comparison with the measured deuterium

concentrations in lateral subsurface flow captured by the trench. We used a constant measured pre-irrigation lateral subsurface flow rate of 30 L h$^{-1}$ (as Graham et al. (2010) ) with a $\delta^2$H signal of -75 ‰ (old water) during the whole sprinkler experiment, and mixed this with the simulated lateral subsurface flow and simulated $\delta^2$H signal at the trench by *Hillvi*. The source of the pre-irrigation lateral subsurface flow is a groundwater seep (well A01), and we refer to this water as deep groundwater in the remaining part of this paper. The simple mixing model also allowed us to compare the simulated $\delta^2$H

signal at the trench by *Hillvi* (mixing without deep groundwater) and the simulated $\delta^2$H signal at the trench as a result of mixing with deep groundwater. Initial state conditions at the start of the *Hillvi* simulations (DOY 208) were estimated by simulating drainage for five days without rainfall input prior to DOY 208, with the best parameter set of the model that was fitted to runoff and mass flux from earlier modeling work at the same site by McGuire and McDonnell (2007) and an estimated bedrock leakage of 0.001 mm hr$^{-1}$ (Graham et al. (2010)). The relative water content in the unsaturated zone five

days prior to DOY 208 was changed manually, until the average simulated water content in the unsaturated zone matched the estimated average water content calculated from the WCR measurements in the three soil pits at DOY 208 and simulated lateral subsurface flow from the trench matched the measured lateral subsurface flow during the first hour of response to the irrigation. We used the WCR measurements for the estimation of the average water content in the unsaturated zone, because these were calibrated using soil cores that were obtained from WS10 and a a number of other locations at the H. J. Andrews

(G. W. Moore et al., unpublished data, 2003; Czarnomski et al., 2005;).

For the parameterization of the model, we used a similar approach as McGuire et al. (2007). After the set-up of the initial state conditions, we performed a Monte Carlo search (5272 runs) over typical parameter ranges based on earlier modeling work and field data. The model performance for lateral subsurface flow was assessed with the Nash Sutcliffe efficiency (*NSE*) (Nash and Sutcliffe, 1970) and the Relative Volume Error (*RVE*), which evaluates the long-term volumetric error.



The *NSE* was used for the δ²H breakthrough in lateral subsurface flow. Behavioral parameter sets for lateral subsurface flow were defined as having a *NSE* value, equal to at least 90% of the highest value obtained during the Monte-Carlo search, and a *RVE* smaller or equal to 0.05. Behavioral parameter sets for the δ²H breakthrough in lateral subsurface flow were defined as having a *NSE* value, equal to at least 80% of the highest value obtained during the Monte-Carlo search. The efficiency of behavioral parameter sets for lateral subsurface flow was defined by weighting *NSE* and *RVE* equally:

$$E_{lateral\ ssf} = 0.5 * NSE_{lateral\ ssf} + 0.5 * (1 - |RVE|) \tag{3}$$

The efficiency of behavioral parameter sets for the δ²H breakthrough in lateral subsurface flow was defined by weighting $NSE_{lateral\ ssf}$, $NSE_{\delta^2 H}$ and *RVE* as follows:

$$E_{\delta^2 H} = 0.25 * NSE_{lateral\ ssf} + 0.25 * (1 - |RVE|) + 0.5 * NSE_{\delta^2 H} \tag{4}$$

The best behavioral parameter set based on $E_{lateral\ ssf}$, and the best behavioral parameter set based on $E_{\delta^2 H}$, were defined as Model 1 and Model 2, respectively. In line with Seibert and McDonnell (2002) and McGuire et al. (2007), we defined uncertainty, expressed as a percentage, as the difference between the 0.9 and 0.1 percentile of the behavioral parameter values divided by the median parameter value..

The original model is written in the IDL development environment (Weiler and McDonnell, 2004) and was converted to Python for our analysis. For the visualization of spatial results of subsurface flow and tracer transport the Mayavi package, a tool for 3D Scientific Data Visualization and Plotting was used.

## 3.6 Analysis of Hillvi model results

After calibrating the *Hillvi* model we used Model 2 to examine the spatial distribution of δ²H concentrations in the unsaturated and saturated zone, and the spatial distribution of recharge fluxes $[M^3 T^{-1}]$ to the saturated zone and lateral subsurface flow $[M^3 T^{-1}]$ during different time slices of the sprinkler experiment. δ²H concentration in the unsaturated zone was determined by:

$$c_{un}(t) = \frac{M_{un}(t)}{S_{un}(t)} \tag{5}$$

where $c_{un}(t)$ is the δ²H concentration in the unsaturated zone of a grid cell at time $t$, $M_{un}(t)$ is the δ²H mass in the unsaturated zone of a grid cell, and $S_{un}(t)$ is the water storage in the unsaturated zone. δ²H concentration in the saturated zone was determined by:

$$c_{sat}(t) = \frac{M_{sat}(t)}{w(t)\ n} \tag{6}$$

where $c_{sat}(t)$ is the δ²H concentration in the saturated zone of a grid cell at time $t$, $M_{sat}(t)$ is the δ²H mass in the saturated zone of a grid cell, $w$ is the water table and $n$ is the total porosity. Lateral subsurface flow for each grid cell was calculated by taking the sum of the eight lateral subsurface flow fluxes leaving that grid cell, whereby negative lateral flow fluxes caused by a negative water table gradient were ignored.



Recharge flux for each grid cell was determined by multiplying the recharge rate $[MT^{-1}]$ by the area of the grid cell. For the analysis, we considered the relative values (0-1) of lateral subsurface flow, recharge flux, $c_{un}$ and $c_{sat}$, at different time slices during the sprinkler experiment. The relative values of these variables were calculated by determining the maximum grid cell value of each variable for all the time slices considered, and dividing the grid cell values by the maximum grid cell value.

Additionally, we used Model 2 to examine the characteristics of vertical recharge and lateral subsurface flow by determining the ratio of lateral subsurface flow and vertical recharge fluxes $[M^3T^{-1}]$. For this analysis, grid cells that received irrigation water were included and recharge velocities smaller than 0.5 mm h⁻¹ were excluded from the analysis to prevent very high ratios as a consequence of small recharge velocities. For each grid cell, the ratio was calculated for mean lateral subsurface flow flux (mean of the eight lateral subsurface flow fluxes leaving the grid cell, excluding zero lateral subsurface flow fluxes) and for maximum lateral subsurface flow flux (maximum of the eight lateral subsurface flow fluxes leaving the grid cell). The spatial distribution of the two ratios was examined at different time slices during the sprinkler experiment. Also, we determined the average dynamics of these ratios by calculating the average of the ratio of lateral subsurface flow flux (mean and max) and recharge flux grid for each model time step.

Finally, we examined with Model 2 simulated exit time and residence time distributions. The exit time distribution is defined as the time elapsed between the entrance of a water particle within a control volume $V$ and its exit through any boundary of the control volume (Botter et al., 2010; Rinaldo et al., 2011). We determined exit time distributions of the labeled sprinkler water for evapotranspiration, recharge, seepage and lateral subsurface flow (captured at trench and total). Because the labeled sprinkler water was applied as a narrow pulse (with respect to the duration of the sprinkler experiment), the exit time distribution is given by:

$$ETD(t) = \frac{C_F(t)}{\int_0^\infty C_F(t)dt} = M_F(t)/M_{out} \tag{7}$$

where $ETD(t)$ is the exit time distribution, $C_F$ is the outflow flux concentration, $M_F(t)$ is the simulated mass flux that exits $V$ and $M_{out}$ the total mass of tracer recovered at the exit boundary, such that the integral of the $ETD(t)$ is unity. The residence time distribution for the storage zone (unsaturated and saturated zones) is given by:

$$RTD_s(t) = M_s(t)/M_{in} \tag{8}$$

where $RTD_s(t)$ is the residence time distribution for the storage zone and $M_s(t)$ is the total mass of tracer in the unsaturated and saturated zone, and $M_{in}$ is the total mass of tracer at the input boundary.

## 4 Results

### 4.1 Flow response to sprinkling

A detectable increase in lateral subsurface flow was observed within an hour of the start of irrigation on DOY 208 (Figure 2). On DOY 210, lateral subsurface flow almost decreased to irrigation discharge levels before the irrigation started, because





on midnight, DOY 210, for 9 h irrigation turned off. Lateral subsurface flow increased within 6 days from an average rate of 30 L h$^{-1}$ prior to irrigation, to a steady-state average rate of 211 L h$^{-1}$. Until DOY 226, when a small sprinkler malfunction occurred, steady-state discharge was maintained. After DOY 228, a series of sprinkler malfunctions resulted in a higher discharge than the steady-state rate. On DOY 232 irrigation was terminated. A distinct diel pattern was evident in lateral

subsurface flow before, during and after the sprinkler experiment.

Soil moisture responded rapidly to irrigation (Figure 2). Within 5-6 days after irrigation (DOY 213-214) soil moisture reached steady-state. . Until DOY 228, steady-state conditions prevailed, since on DOY 228 the first of the sprinkler malfunctions produced a rise in soil moisture. On DOY 232 irrigation was turned off, and as a result the soil profile drained rapidly during the first 8–12 h, which was followed by a slower, more stable drainage during the remainder of the sprinkler

experiment. Soil moisture in the upper 0.30 m started to increase after 30 min of irrigation, soil moisture at 0.30-0.60 m started to increase after 45 min, and soil moisture sensors below 0.60 m started to increase after 100 min. Estimated median wetting front celerities from the soil moisture measurements ranged from 43 to 150 mm h$^{-1}$ (Table 1), with highest estimated median wetting front celerities for the upper 0.60 m.

Tensiometers at 0.30 m also responded quickly to irrigation (Figure 2). Steady-state conditions were reached within 2 days

after irrigation (DOY 210), with matric potential values varying between 0.85 and -0.10 kPa (diel fluctuations). Tensiometers at 0.70 m reached steady-state conditions 4 days after irrigation (DOY 212), with matric potential varying between 0 and -0.50 kPa (diel fluctuations). The tensiometer at 1.0 m reached steady-state conditions 5 days after irrigation (DOY 213), with matric potential values varying between 1.5 and 0.80 kPa (diel fluctuations). Untile DOY 228 steady-state conditions prevailed, when matric potential increased, caused by the first sprinkler malfunction. Estimated median wetting

front celerities from the soil matric potential measurements ranged from 10 to 116 mm h$^{-1}$, with highest estimated median wetting front celerities for the upper 0.30 m (Table 1).

We found a similar rapid response to irrigation in groundwater wells A05 and E04, 1.2 and 4.3 h after irrigation respectively. Estimated wetting front celerities were 377 and 87 mm h$^{-1}$ for well A05 and E04 respectively (Table 1). Sampling of the groundwater wells during the sprinkler experiment disturbed the groundwater dynamics to such extent that evaluation of

steady-state conditions was not possible.

### 4.2 Deuterium breakthrough in soil, ground and lateral subsurface water

Within 4 h of $\delta^2$H application labeled $\delta^2$H water was observed in lateral subsurface flow and $\delta^2$H peaked at approximately 20 h (Figure 3). Deep groundwater exfiltrating from the seep at well A01 did not show any response in $\delta^2$H during the entire sprinkler experiment (Figure 4). In contrast, $\delta^2$H in the transient water table increased rapidly (Figure 4, well A05, E04), with

peak values within 2 and 3 days after application of $\delta^2$H labeled water. All soil lysimeters revealed large $\delta^2$H increases. The response to the $\delta^2$H input was first observed at the shallow lysimeters at 0.3 m depth, while and the observed $\delta^2$H peak in lysimeters at 0.7 and 1.1 m depth followed the $\delta^2$H peak at 0.3 m depth (Table 2 and Figure 4).



For all locations, the $\delta^2$H peak arrival time ranged from 1 to 8 days and lagged with depth. Based on these $\delta^2$H peak arrival times, the average vertical tracer velocity ranged from 6 to 17 mm h$^{-1}$. The average vertical traces velocity at well E04 was about 2-3 times faster than the average vertical velocity of the deep lysimeters, and for this location the $\delta^2$H peak arrival time did not lag with depth (Table 2).

## 4.3 Mixing and flowpaths at the hillslope scale

For water collected from the trench exiting the hillslope, the maximum event water fraction was 26%, calculated by the $\delta^2$H-based two- component mixing model and occurred at DOY 209, 20 h after the $\delta^2$H application but prior to the addition of non-labeled sprinkler water (Figure 5). After DOY 209, the event water fraction gradually decreased to ~1% on DOY 234. When the deep groundwater contribution with an old water isotopic signature in lateral subsurface flow was excluded, the $\delta^2$H-based two component mixing model showed a maximum event water fraction of 37%.

Soil water $\delta^2$H, sampled by suction lysimeters, peaked 1 day after the $\delta^2$H application (DOY 209), at 0.3 m depth, which corresponded with nearly saturated conditions at the lower and middle slope position tensiometers at the same depth. We defined the arrival time of nearly saturated conditions as the first three successive measurements with $\psi$ < 0.1 kPa and occurred at the lower and middle slope position tensiometers at 0.3 m depth 28 h (DOY 209.41) and 13 h (DOY 208.76) after irrigation respectively. Also, soil moisture at 0.15-0.3 m and 0.3-0.6 m depth attained nearly steady-state water contents 18 h (DOY 208.96) and 21 h (DOY 209.11) after irrigation respectively. This suggests that shallow soil water was the main source of the observed $\delta^2$H peak (DOY 209.42) in lateral subsurface flow measured at the trench.. About 4 days after irrigation, nearly saturated conditions occurred at 0.7 m depth, at the lower and middle slope position on DOY 212.18 and 211.86 respectively. Soil moisture at 0.9 and 1.2 m depth attained nearly steady-state water contents 3 days after irrigation on DOY 211.09 and 211.43, respectively. The $\delta^2$H peak at well E04 and deeper tension lysimeters was observed 3-4 days (DOY 212-213) and 3 days (DOY 211.74) after the $\delta^2$H application, respectively. Thus, deeper soil water became important 3 days after irrigation (DOY 212).

## 4.4 Hillvi modeling results

The best parameter set based only on lateral subsurface flow is presented in Table 3 (Model 1) with NSE of 0.96 and a RVE of 0.0004 for lateral subsurface flow, and a NSE of 0.82 for the $\delta^2$H breakthrough in lateral subsurface flow. Only one parameter, hydraulic conductivity of bedrock ( $k_b$ ), appears to be identifiable based on relative frequency of the 796 behavioral parameter sets (Figure 6). The best parameter set based on both lateral subsurface flow and the $\delta^2$H breakthrough in lateral subsurface flow (Model 2, Table 3) resulted in a NSE of 0.94 and a RVE of 0.0127 for lateral subsurface flow, and a NSE of 0.93 for the $\delta^2$H breakthrough in lateral subsurface flow. The optimum values of most parameters of Model 2 were different from Model 1, only parameters $k_b$, $f$ and $\beta$ had similar optimum values (Table 3). The relative frequency of the



308 behavioral parameter sets based on lateral subsurface flow and $\delta^2$H (Figure 6) indicate that parameters $k_b$, $n$, $n_0$ and $*n_{eff}$ were identifiable. The parameter $*n_{eff}$ was only included in Model 2, since this parameter is used to describe the available pore volume for mass transfer. In addition, the relative uncertainty of these parameters for Model 2 was lower compared to Model 1.

While Models 1 and 2 both produced similar results for lateral subsurface flow, the results for the $\delta^2$H breakthrough in lateral subsurface flow were different (Figure 3). Model 1 underestimates the first $\delta^2$H peak considerably; however based on the objective criterion for the $\delta^2$H breakthrough, we did not reject Model 1 as a behavioral parameter set. Figure 3 also presents the $\delta^2$H breakthrough in lateral subsurface flow without mixing with deep groundwater. Based on the simple mixing model between deep groundwater with an old water signal and the simulated $\delta^2$H in lateral subsurface flow, the simulated

maximum event water fraction by Model 2 for mixing with and mixing without deep groundwater was 0.28 and 0.40 (Table 4), respectively, at DOY 209, about 25 h after the $\delta^2$H application. For total lateral subsurface flow (including flow that bypassed the trench but excluding deep groundwater), the simulated maximum event water fraction was 0.38 (Table 4). Based on the 308 behavioral parameter sets, the highest maximum event fraction was 0.60 for lateral subsurface flow from the trench and total lateral subsurface flow, both excluding the contribution of deep groundwater (Table 4).

Mass recovery for total water applied during the sprinkler experiment and $\delta^2$H applied during the labeled pulse was highest for total lateral subsurface flow and lateral subsurface flow collected at the trench, as well as for Model 2 and the 308 behavioral parameter sets (Table 4). Total bedrock seepage $\delta^2$H mass flux was about half of total lateral subsurface flow $\delta^2$H mass flux for Model 2. About 10% of the total $\delta^2$H input of the labeled pulse remained in the unsaturated zone as a result of the inclusion of the effective porosity parameter in Model 2. Of total water applied during the $\delta^2$H pulse, only 2% remained

in the unsaturated zone (Table 5). This resulted in higher fractions of total water applied during the sprinkler experiment for total lateral subsurface flow and lateral subsurface flow collected at the trench, compared to fractions of total $\delta^2$H input of the labeled pulse for total lateral subsurface flow and lateral subsurface flow collected at the trench.

The simulated median residence time of $\delta^2$H in the storage zone of the hillslope was 7.4 (99.1) days (Figure 7) during the field experiment, where the value between parentheses refers to the translation of field experiment days to natural conditions.

Although the simulated median residence time of $\delta^2$H was not much longer than the median travel time of total lateral subsurface flow (7.2 (95.6) days), about 10% of the total mass input of the labeled sprinkler water remained in the unsaturated zone (as a consequence of the inclusion of the effective porosity parameter), suggesting that the simulated median residence time of 7.4 (99.1) days in the storage zone should be interpreted as a minimum. The simulated median travel time of $\delta^2$H in lateral subsurface flow (5.3 (69.5) days) captured at the trench was lower than the simulated median

travel time of $\delta^2$H in total lateral subsurface flow (7.2 (95.6) days). This is probably caused by the spatial variation of soil depth at the hillslope, whereby the trench is fed by water from more shallow soils compared to total lateral subsurface flow. The similar exit time distributions (CDF) of total lateral subsurface flow, seepage and recharge, and the different exit time distribution of lateral subsurface flow at the trench, indicate that water captured at the trench was not representative for the



hydrological and mass transport behavior of the entire hillslope domain that generated total lateral subsurface flow. The exit time distribution of evaporation was characterized by a heavier tale, compared to the other exit time distributions.

Figure 8 presents the relative concentration of $\delta^2$H in the unsaturated and saturated zone and the relative fluxes of recharge and subsurface flow at different time slices simulated by Model 2, to illustrate the spatial dynamics of these variables during

the sprinkler experiment. Presenting the relative concentration of $\delta^2$H in the unsaturated and saturated zone and the relative flux of recharge and subsurface flow separately preserves more information than showing relative mass fluxes of $\delta^2$H. $\delta^2$H in lateral subsurface flow around the $\delta^2$H peak, T=10 and T=20 h after the application of labeled sprinkler water (DOY 209), was mainly derived from the lower part of the hillslope, characterized by shallow soil depths. This estimate agreed with the tensiometer and soil moisture observations mentioned earlier. $\delta^2$H in the unsaturated zone varies spatially at T=10 and T=20

h, because of soil depth variation, with deeper soils upslope and thus a larger mixing volume with pre-event soil water, resulting in lower $\delta^2$H concentrations. A similar pattern for relative recharge fluxes emerged because of soil depth variation, with low relative recharge fluxes in deep soil upslope, until at least 64 h after the application of labeled sprinkler water. The spatial pattern of relative $\delta^2$H concentrations in the saturated zone are similar to the spatial pattern of relative $\delta^2$H concentrations in the unsaturated zone, except that the relative $\delta^2$H concentrations in the saturated zone show a more diffuse

pattern because of a 'mechanical (convective) dispersion' caused by partitioning of outflow from each grid cell. T = 20 and 30 h after the application of labeled sprinkler water (DOY 209) the relative $\delta^2$H concentration in the unsaturated and saturated zone decreased because of dilution with non-labeled sprinkler water while relative fluxes of subsurface flow and recharge increased.

Figure 9a and 9b demonstrate the characteristics of simulated vertical recharge flow and lateral subsurface flow during the

sprinkler experiment by plotting the ratio of lateral subsurface flow and vertical recharge fluxes for the 308 parameter sets of Model 2. During the sprinkler experiment mean and maximum lateral subsurface flow fluxes were mostly larger than the vertical recharge fluxes. While Model 2 showed ratios smaller than one during DOY 208 and 209, a large number of simulations showed higher ratios than one (e.g. 65 and 125 simulations were higher than one on DOY 209.5 for mean and maximum lateral subsurface flow, respectively). Figure 10 presents the spatial variation of the ratio of vertical recharge and

mean and maximum lateral subsurface flow fluxes simulated by Model 2 at different time slices during the sprinkler experiment. During the initial $\delta^2$H breakthrough in lateral subsurface flow (T=10 and 20 h) the ratio of lateral subsurface flow and vertical recharge fluxes was around one near the trench and values decreased upslope. After T=10 h the ratio of lateral subsurface flow and vertical recharge fluxes increased to a maximum value of about three and six for mean and maximum lateral subsurface flow, respectively. In addition, after T=20 h higher values for the ratio of lateral subsurface flow

and vertical recharge fluxes also occurred upslope.





## 5. Discussion

### 5.1 Process understanding through the Hillvi model

We performed a two-step model calibration, first on lateral subsurface flow data alone and then on lateral subsurface flow data and the deuterium breakthrough in lateral subsurface flow. As found in the modeling study of McGuire et al. (2007), the

inclusion of tracer information improved parameter identification and thus provided further insight into the processes that control hillslope scale water flux and the rapid mobilization of stored, pre-event (old) water to the stream.

Effective porosity ($n_{eff}$) was an important and identifiable parameter (Figure 5) for the simulation of the deuterium breakthrough in lateral subsurface flow. This parameter represents a dual porosity system (e.g. Buttle and Sami, 1990; Corapcioglu and Wang, 1999; Stephens et al., 1998; Rasmussen et al., 2000) with a mobile and immobile water domain and

the calibration result for this parameter suggests that the mixing volume of deuterium was controlled by an immobile soil water fraction. A dual porosity system is consistent with the study of Brooks et al. (2010) in the same small watershed: based on water-isotope data, they found soil water that is tightly bound (immobile) within small soil pores remains in the soil or leaves the soil through transpiration by trees but is does not contribute to matrix flow, and is disconnected from mobile or stream water.. Although a dual porosity system is consistent with the study of Brooks et al. (2010), we cannot compare the

results of this study directly to Brooks et al. (2010), since the difference in timescale between both studies. The bypass term was somewhat less identifiable compared to previous work by McGuire et al. (2007). However, model calibrations on lateral subsurface flow and $\delta^2H$ showed that higher bypass values produced better simulations (not shown); this is in contrast with McGuire et al. (2007) who found that smaller bypass values to produced better simulations. Our finding though is consistent with dye staining experiments at a nearby site (McGuire et al., 2007), other studies at this site (McGuire and McDonnell,

2010; van Verseveld et al., 2008) and observations published in other studies (Radulovich et al., 1992; Hornberger et al., 1990; van Stiphout et al., 1987; Jackson et al., 2016) that all showed the importance of bypass flow. Additionally, our finding is in agreement with higher observed average vertical velocity at well E04 compared to the deep lysimeters during the sprinkler experiment.

Furthermore, the hydraulic conductivity of bedrock was an important and identifiable parameter for the simulation of lateral

subsurface flow and the deuterium breakthrough. Graham et al. (2010) showed that deep seepage at the WS10 hillslope scale was 48 ± 35% of applied water during the same irrigation period (calculated after 10 days of drainage), based on water balance calculations. Our modeling results (Model 2) showed that bedrock seepage accounted for 24% and 26% based on total applied water and $\delta^2H$ mass of the labeled pulse respectively. Graham et al. (2010) hypothesized that water that bypassed the trench system was at most 10% of the measured lateral subsurface flow while Model 2 simulated that 70% of

lateral subsurface flow was captured by the trench and 30% bypassed the trench. As a consequence of the lower bypass estimate by Graham et al. (2010), they estimated higher deep seepage fractions. However, our simulated bedrock seepage is consistent with the estimated deep seepage at the catchment scale by Graham et al. (2010) that was on average approximately 21% of precipitation at steady-state. Our modeling approach of bedrock seepage (with no later exfiltration of



bedrock water) is largely in agreement with the study of Gabrielli et al. (2012) at the same hillslope. Gabrielli et al. (2012) drilled into the underlying breccia at seven locations and found no indication that bedrock groundwater moved upward and contributed this way to lateral subsurface stormflow. However, they found that water from very shallow substantially fractured bedrock near the soil-bedrock interface contributed to lateral subsurface stormflow. Their interpretation was that the source of this water consisted mainly of infiltrating rainwater than mixed with soil water. Furthermore, their perceptual model of water flow through the WS10 hillslope showed a deeper seepage component returning as baseflow, similar to our definition of deep groundwater (well A01 and baseflow prior to the start of the sprinkler experiment).

**5.2 Celerities and velocities at the hillslope scale**

The average vertical flow velocities of $\delta^2H$ were generally much slower than celerities estimated from wetting front arrival times from soil moisture, matric potential and groundwater height measurements. For example, median celerities at 0.60-0.70 m depth were three and twenty times faster, for TDR and tensiometer measurements respectively, than average vertical flow velocities (7 mm h$^{-1}$) at 0.70 m depth. Torres et al. (1998) found on a study hillslope with comparable high permeability and porosity soils that at the onset of a sprinkler experiment, a pressure head signal advanced on average fifteen times faster than estimated water and wetting front celerities. They attributed this to the advancement of a pressure wave instead of advective flow of new water. Following recommencement of irrigation, Jackson et al. (2016) observed during their multitracer hillslope irrigation experiment a fast trench flow and piezometer response indicating a pressure wave celerity much faster than observed dye tracer velocities. Rasmussen et al. (2000) presented parametric expressions for the celerity, which predicted pressure wave travel times two to fifteen times faster than the tracer velocity for their short-duration fluid irrigation experiments with intact saprolite columns. How the perturbation (irrigation) was exactly transmitted through the flow domain at our site remains unclear. For example, the celerity at our site could have been produced by a pressure wave response that gave rise to rapid soil water redistribution (as per Torres, 2002). On the other hand, McGuire and McDonnell (2010) indicated that advective preferential flow transport was the most plausible mechanism for observed rapid soil moisture responses in the unsaturated zone at this study site.

While the precise mechanism of disturbance transmittance remains unclear at our study site, our modeling results are consistent with celerities that were much faster than average vertical flow velocities. The $\delta^2H$ breakthrough curve for lateral subsurface flow was charactized by an early breakthrough and a long tail. This breakthrough shape indicates a subsurface system where mixing is controlled by an immobile soil fraction. This is supported by the importance of the effective porosity model parameter for modeling the $\delta^2H$ breakthrough curve. The observed peak $\delta^2H$ value of about +20 ‰ in lateral subsurface flow differed substantially from the labeled irrigation water with an average signal of +285 ‰. In *HillVi*, complete mixing of $\delta^2H$ with available unsaturated and saturated stores was assumed, but the actual transport of $\delta^2H$ was only associated with a portion of the total available porosity (effective porosity). The effect of the effective porosity model parameter was to decouple the hydrologic response (celerity) from the velocity of water transport, consistent with calculated faster celerities than average vertical flow velocities from field observations.



### 5.3 Residence and travel time distributions

The estimated simulated median travel time for total lateral subsurface flow during natural conditions (95.6 days) was in the same range as reported by McGuire et al. (2007) for the same site. McGuire et al. (2007) did for example report a simulated mean travel time of 92 days during steady-state conditions calculated with their model that was calibrated to runoff and mass

flux of tracer.. They observed that these simulated distributions were considerable younger than estimates that use measured stable isotopes in runoff (McGuire et al., 2005). One of the explanations, besides that observed stable isotope signatures reflect largely baseflow conditions, was that the mixing volume of model did not include bedrock contributions. While we did not explicitly model bedrock contributions during the sprinkler experiment, we did incorporate the contribution of deep groundwater, resulting in a lower simulated maximum event water contribution and an older simulated travel time

distribution, compared to the situation without contribution of deep groundwater.

The simulated median residence time was similar to simulated median travel times, a somewhat expected result since we know from theoretical work of Rinaldo et al. (2011) that the wetter the catchment the closer the median residence and travel time distributions become—our sprinkler experiment was characterized by generally wet conditions. On the other hand, the median residence time in the storage zone should be considered a minimum value, because about 10% of the total mass input

of the labeled sprinkler water remained in the unsaturated zone. Botter et al. (2010) showed theoretically that under wet conditions the travel time probability density function (PDF) and evapotranspiration PDF are only slightly different, and this was explained by relatively high soil moisture contents and thus high efficiency of the transpiration processes. Indeed, generally, the simulated evaporation PDF by *HillVi* was similar to the other simulated travel time distributions with an early time peak. While similar, the evaporation PDF showed a more pronounced diel fluctuation, heavier tail and more weighting

at earlier times than the other travel time distributions. This difference is probably caused by a combination of evapotranspiration fluxes (pronounced diel fluctuations), effective porosity and low subsurface flow velocities (heavier tale) and simply because the unsaturated zone firstly responds to irrigation (more weighting at earlier times). The exit time CDFs of total lateral subsurface flow, and lateral subsurface flow captured at the trench, illustrated clearly that the exit time CDF of lateral subsurface flow captured at the trench showed more weighting at earlier times.

### 5.4 What controlled the tracer breakthrough curve at the hillslope scale?

Besides a dual porosity system (effective porosity), the constant contribution of deep groundwater further reduced the $\delta^2H$ signal in lateral subsurface flow. Both processes caused an attenuation of the observed $\delta^2H$ spike in irrigation water and produced a small contribution of event water. These processes explain how stored, pre-event water quickly mobilizes. However, this explanation is not in line with results of Anderson et al. (1997) who concluded that old water rapidly

mobilizes because of old water displacement (plug flow) in the unsaturated zone mixing with lateral subsurface flow originating from the bedrock. Our explanation is more in agreement with the study of Collins et al. (2000) who did not find evidence for new water that completely displaces old water. Similarly, Kienzler and Naef (2007) inter-compared sprinkler



experiments at different locations, and found that sites where large pre-event (old) water fractions were observed, were characterized by subsurface flow, that was supplied indirectly by preferential flow paths from saturated soil parts. Their study did not find support for pre-event and event water that completely and instantaneously mixed. Instead, they found that pre-event water constantly emanated from small pores to large pores during saturation of the soil. Our modeling results of

the $\delta^2$H breakthrough curve in lateral subsurface flow point to a similar process at our site whereby event water that moved to the immobile pool (i.e. dead-end pores) during the start of the sprinkler experiment was gradually released during the rest of the sprinkler experiment.

The two component mixing model based on measured $\delta^2$H showed a small contribution of event water (maximum of 26% with deep groundwater contribution) to lateral subsurface flow. We identified two main hillslope flowpaths: vertical water

movement through the unsaturated zone and lateral subsurface flow that traveled quickly downslope. $\delta^2$H peaks in lysimeter water and transient groundwater showed that vertical flow moved to depth sequentially. Additionally, the soil moisture and tensiometer measurements showed a sequential response with depth. Well E04 was the exception because it was characterized by an earlier arrival time of the $\delta^2$H peak than the deep soil. This indicates preferential flow paths that by-passed the deep soil at this location. Vertical bypass flow was an essential process to include in *Hillvi* in order to capture the

lateral subsurface flow and the $\delta^2$H breakthrough in lateral subsurface flow dynamics during the sprinkler experiment.

Through the combination of isotopic ($\delta^2$H) and tensiometer data, this study showed that the source of $\delta^2$H peak in lateral subsurface flow was primarily from shallow soil water nearby the trench. The soil depth at this site varies between 0.1-0.3 m nearby the hillslope trench to about a soil depth of 3 m at the upper end of the sprinkler plot. This soil depth variation in combination with vertical flow though the unsaturated zone and lateral subsurface flow downslope at this site is in agreement

with a shallow soil source of the $\delta^2$H peak in lateral subsurface flow. Also our modeling results are in line with the shallow soil depth source of the $\delta^2$H peak. Since the $\delta^2$H signal in well E04 indicated preferential flow, water may have originated from deeper soils than 0.3 m higher upslope.. The spatial variability of soil depth at our site most likely also played an important role in the generation of the $\delta^2$H breakthrough curve. Several studies, at WS10 (e.g. Sayama and McDonnell, 2009) and elsewhere (e.g. Hopp and McDonnell, 2009) have shown that increases in soil depth lead to a general attenuation

of the hydrological response. Deeper soils increase the vertical travel distance to the soil bedrock interface, have more storage volume and thus increase the mixing reservoir with pre-event water and more slowly reach saturation, resulting generally in a more damped signal of $\delta^2$H in lateral subsurface flow. Figures 8 and 10 clearly show the disconnection of more upslope areas (deeper soils) from the trench in terms of subsurface flow and $\delta^2$H, around the $\delta^2$H peak in lateral subsurface flow.

## 6. Conclusions

This study combined isotopic and internal physical measurements for a 24 day hillslope sprinkler experiment. The steady irrigation rates and controlled conditions avoided issues associated with natural variability in inputs, and enabled the





application of a process-based spatial explicit hydrologic model. This resulted in a mechanistically plausible conceptual model for celerity-velocity differences and the rapid mobilization of old water. Our analyses show that flow paths of water were vertical flow through the unsaturated zone and then strongly lateral in the thin zone above the bedrock for rapid subsurface flow. Celerities estimated from wetting front arrival times were generally much faster than average vertical

velocities of $\delta^2H$. This was consistent with our modeling results that showed that the transport of $\delta^2H$ through the hillslope was controlled by effective porosity, indicating a subsurface system where mixing is controlled by an immobile soil fraction. This was one of the mechanisms that caused the large reduction in the $\delta^2H$ input signal in lateral subsurface flow. Furthermore, deep groundwater caused a further reduction in the $\delta^2H$ input signal, based on calculations with a simple mixing model. Finally, we showed that the soil depth variability at our site played an important role in the generation of the

$\delta^2H$ breakthrough curve. Deeper upslope soils damped the $\delta^2H$ input signal. These processes reduced the observed $\delta^2H$ spike in irrigation water and produced a small contribution of event water, explaining therefore the fast appearance of stored, pre-event water in lateral subsurface flow.

The broader implications of our work showed that modeling exit time and residence time distributions provided added value. Through the modeling we observed that the trench did not represent the total modeled hillslope domain, since lateral

subsurface flow captured at the trench showed more weighting at earlier times. Additionally, the residence (10% $\delta^2H$ remained in the unsaturated zone) and exit time distributions (heavy tail) of $\delta^2H$ in the unsaturated zone, in combination with the effective porosity as an important model parameter, illustrated the important role of the unsaturated zone in controlling mass transport at the hillslope scale. Finally, besides the unsaturated zone, this study revealed bedrock seepage and deep groundwater as important controls on mass transport at the hillslope scale.

**Acknowledgements**

This work was supported through funding from the National Science Foundation (grant DEB 021-8088 to the Long-Term Ecological Research Program at the H. J. Andrews Experimental Forest), Department of Forest Engineering at Oregon State University and the Environmental Protection Agency. This manuscript has been subjected to the Environmental Protection Agency's peer and administrative review, and it has been approved for publication as an EPA document. The views

expressed in this paper are those of the author(s) and do not necessarily reflect the views or policies of the U.S. Environmental Protection Agency. Mention of trade names or commercial products does not constitute endorsement or recommendation for use. We thank Marloes Bakker, Matthew Bergen and John Moreau for providing field assistance. We especially thank the McKenzie River Ranger District for providing irrigation water during the experiment, and Kari O'Connell and Cheryl Friesen for coordinating logistics. We also thank R. D. Harr and D. Ranken for initiating the hillslope

studies at WS10, and K. J. McGuire for re-initiating this site.



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





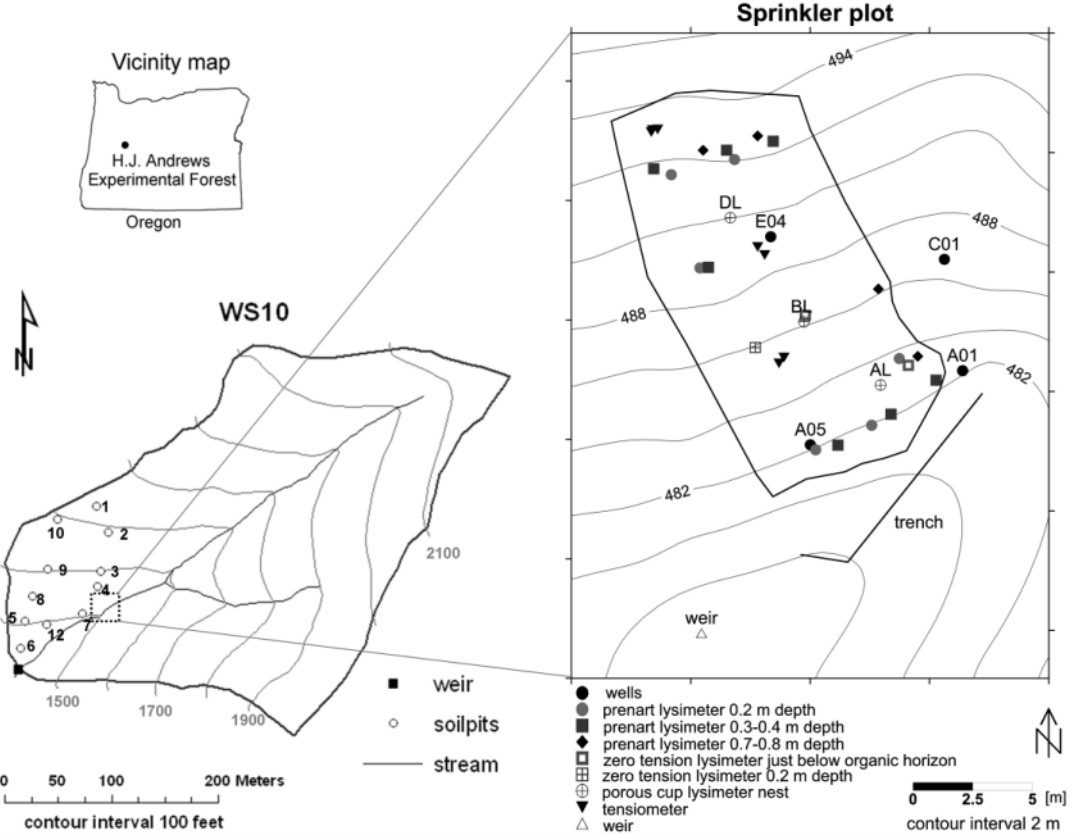

5    **Figure 1: Map of WS10 with soil pits and sprinkler area with instrumentation. See Methods 3.1 for abbreviation explanations.**



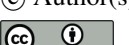

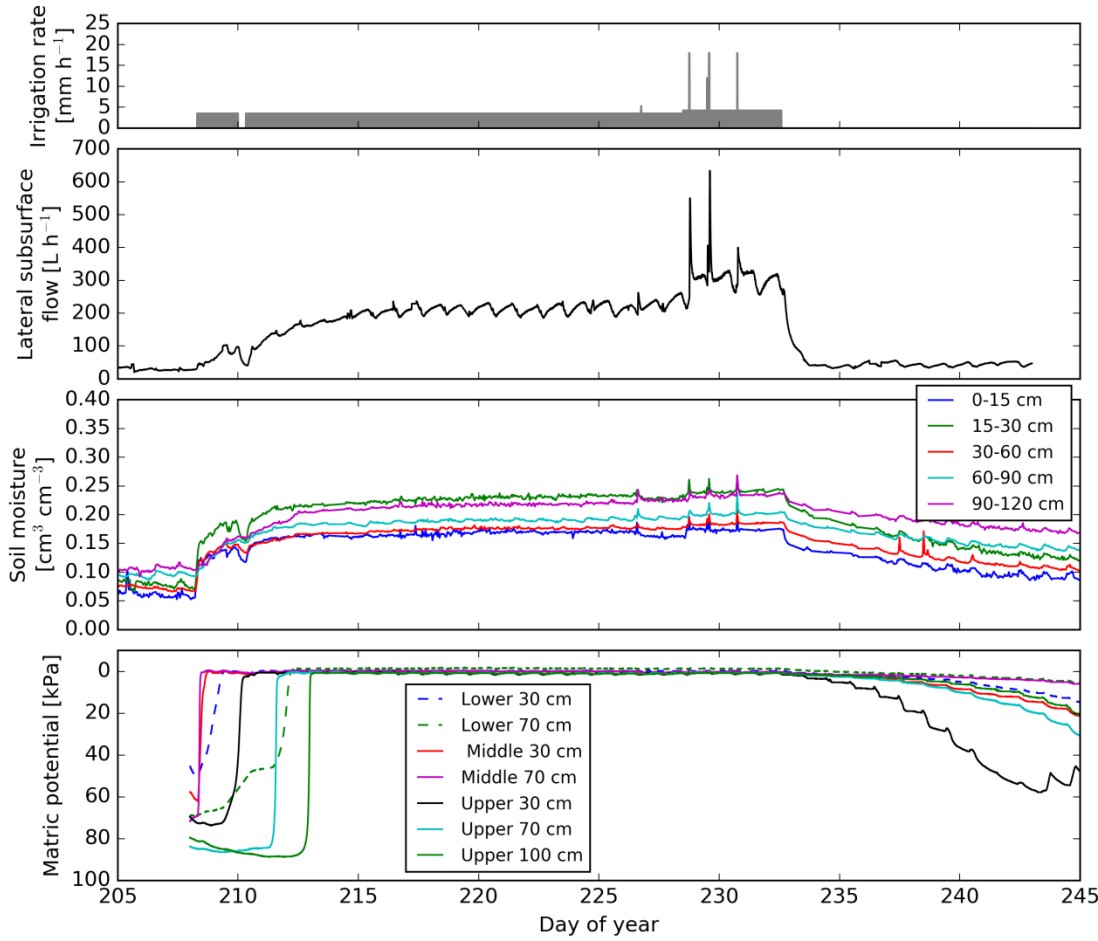

**Figure 2: Temporal dynamics over the experiment of the irrigation rate (top panel), lateral subsurface flow measured at the hillslope trench (second panel), soil moisture measurements averaged for five measurement depths (third panel) and matric potential measurements for the three tensiometer nests (bottom panel).**





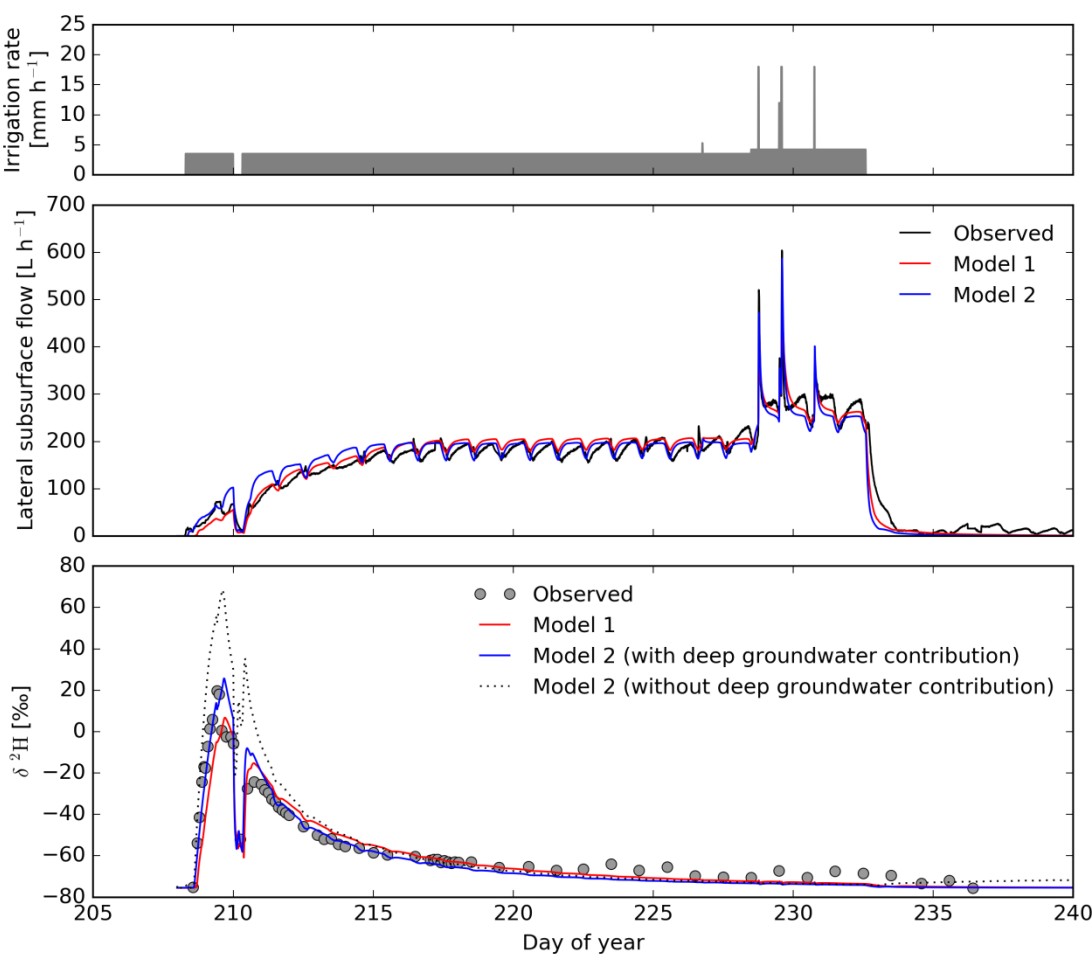

**Figure 3: Observed (at the trench) and simulated lateral subsurface flow and δ²H breakthrough in lateral subsurface flow, for Model 1 and 2.**



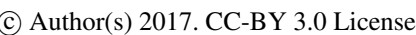

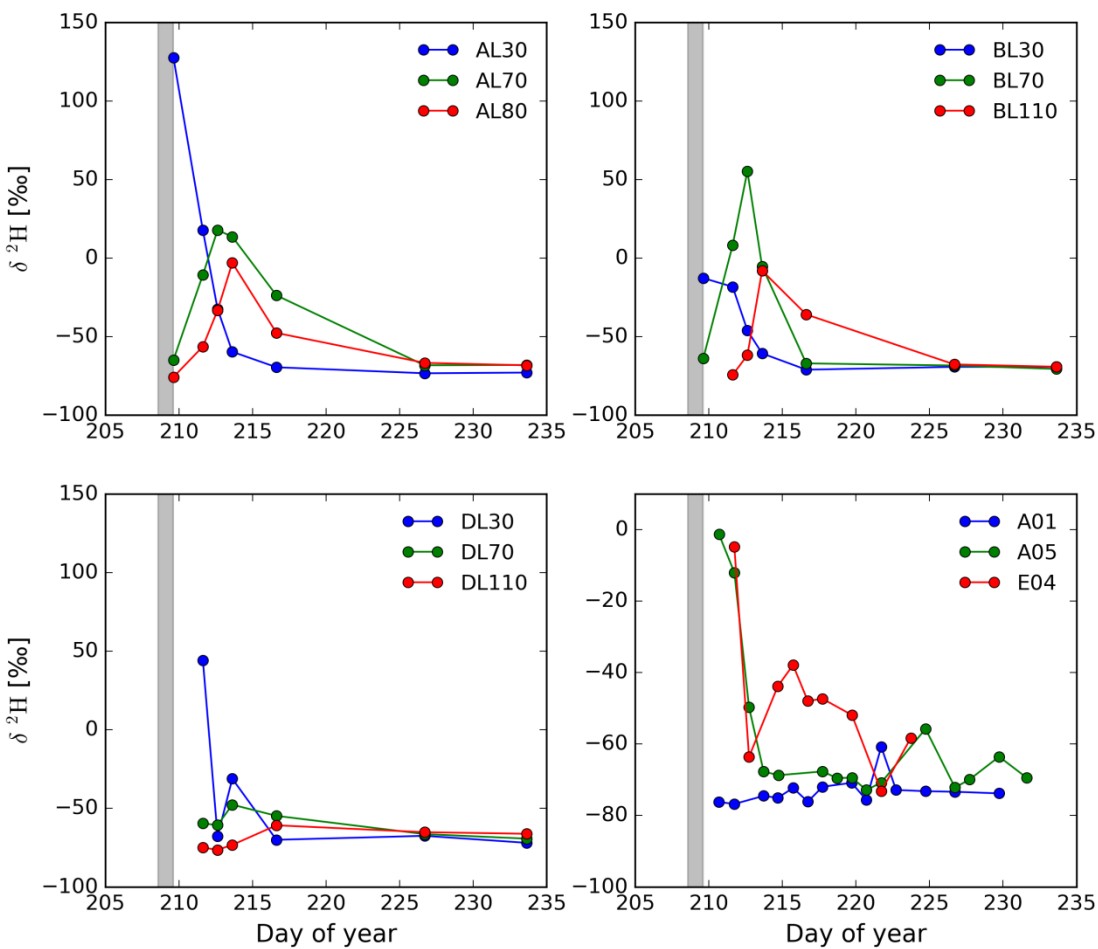

**Figure 4: Soil water δ2H breakthrough curves at tension lysimeter nests AL, BL and DL, and wells A01, A05 and E04. Gray area represents application of labeled irrigation water.**



**Table 1: Median wetting front celerities estimated from different type of measurements (GW=Groundwater, TDR = Time Domain reflectometry and TM = Tensiometer).**

| Depth [m] | Measurement | Number of measurements | Median celerity [mm h$^{-1}$] |
|---|---|---|---|
| 0.15 | TDR | 9 | 110 |
| 0.30 | TDR | 11 | 150 |
| 0.30 | TM | 3 | 120 |
| 0.44 | GW (A05) | 1 | 380 |
| 0.60 | TDR | 13 | 150 |
| 0.70 | TM | 3 | 25 |
| 0.90 | TDR | 11 | 45 |
| 1.00 | TM | 1 | 9.7 |
| 1.20 | TDR | 8 | 43 |
| 1.24 | GW (E04) | 1 | 87 |





**Table 2: Calculated velocities based on timing of $\delta^2H$ peak and mean arrival time of $\delta^2H$ (first normalized moment of the BTC).**

| Location | Peak $\delta^2H$ [‰] | Response time [d] to $\delta^2H$ peak input | Depth [m] | Velocity [mm h$^{-1}$] based on $\delta^2H$ peak |
|---|---|---|---|---|
| A05 | -1.3 | 2 | 0.44 | 9.2 |
| E04 | -5 | 3 | 1.24 | 17 |
| AL30 | 128 | 1 | 0.30 | 13 |
| BL30 | -12.7 | 1 | 0.30 | 13 |
| DL30 | 44.3 | 1 | 0.30 | 13 |
| AL70 | 17.9 | 4 | 0.70 | 7.3 |
| BL70 | 55.3 | 4 | 0.70 | 7.3 |
| DL70 | -47.7 | 5 | 0.70 | 5.8 |
| AL80 | -3 | 5 | 0.80 | 6.7 |
| BL110 | -7.9 | 5 | 1.10 | 9.2 |
| DL110 | -60.8 | 8 | 1.10 | 5.7 |
| ZTL20_1 | 16.2 | 1 | 0.20 | 8.3 |





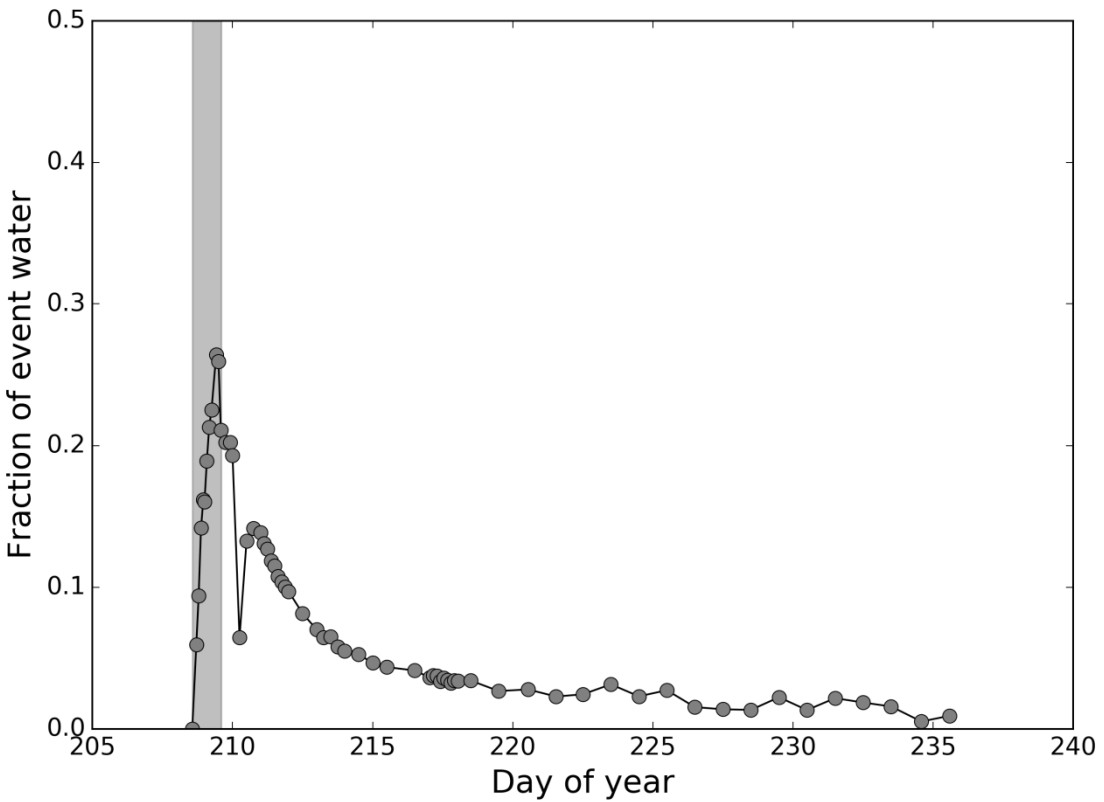

**Figure 5: Fraction of event water captured at the trench during the sprinkler experiment based on a 2-component mixing model. Grey area represents application period of labeled irrigation water.**





**Table 3: Parameter description, ranges (calibration), best parameter sets with uncertainty in parentheses, based on lateral subsurface flow (Model 1) and based on both lateral subsurface flow and $\delta^2H$ (Model 2). Parameter uncertainty is defined as the range between behavioral parameter values of the 0.1 and 0.9 percentiles divided by the median parameter value expressed as a percentage.**

| Parameter | Description | Parameter range | | Model parameters | |
|---|---|---|---|---|---|
| | | Lower limit | Upper limit | Model 1 | Model 2 |
| $n$ | Average soil porosity | 0.42 | 0.56 | 0.48 (22%) | 0.44 (18%) |
| $b$ (m) | Shape factor for drainable porosity function | 1 | 2 | 1.8 (53%) | 1.4 (54%) |
| $n_0$ | Surface drainable porosity | 0.17 | 0.30 | 0.20 (43%) | 0.30 (37%) |
| $f$ (m) | Shape factor for hydraulic conductivity function | 0.5 | 0.8 | 0.72 (36%) | 0.75 (36%) |
| $k_0$ (m h$^{-1}$) | Surface hydraulic conductivity | 4.4 | 9 | 7.7 (57%) | 8.4 (56%) |
| $c$ | Recharge power coefficient | 23 | 114 | 27.0 (102%) | 42.4 (113%) |
| $\beta$ | Bypass power coefficient | 5 | 30 | 26.5 (137%) | 23.6 (145%) |
| *$n_{eff}$ | Effective porosity coefficient | 0.1 | 1 | - | 0.56 (86%) |
| $k_b$ (m h$^{-1}$) | Hydraulic conductivity of bedrock | 0 | 0.002 | 0.00085 (75%) | 0.00095 (45%) |





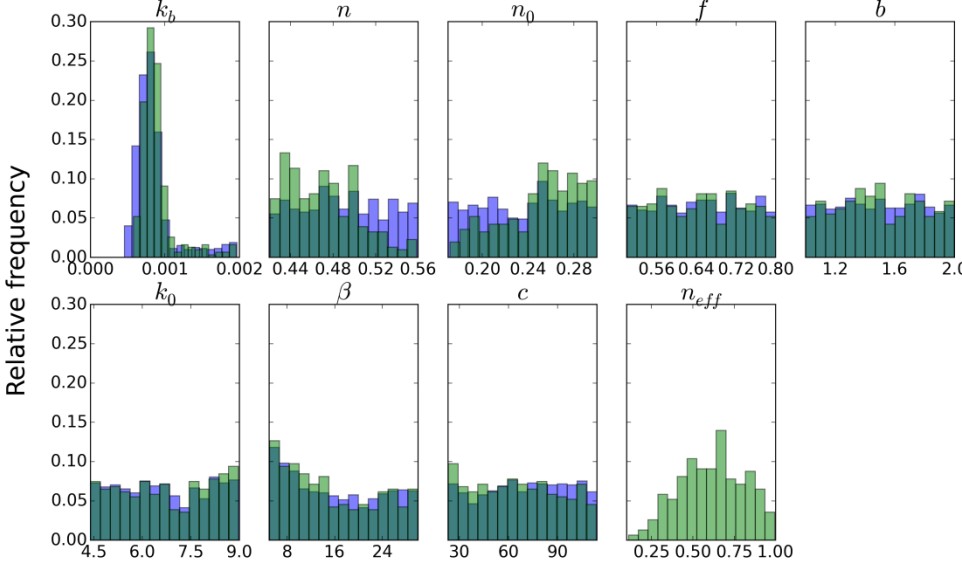

**Figure 6: Relative frequency plots of the 796 behavioral parameter sets (blue color) based on lateral subsurface flow (Model 1), and relative frequency plots of the 308 behavioral parameter sets (green color) based on lateral subsurface flow and δ2H (Model 2).**





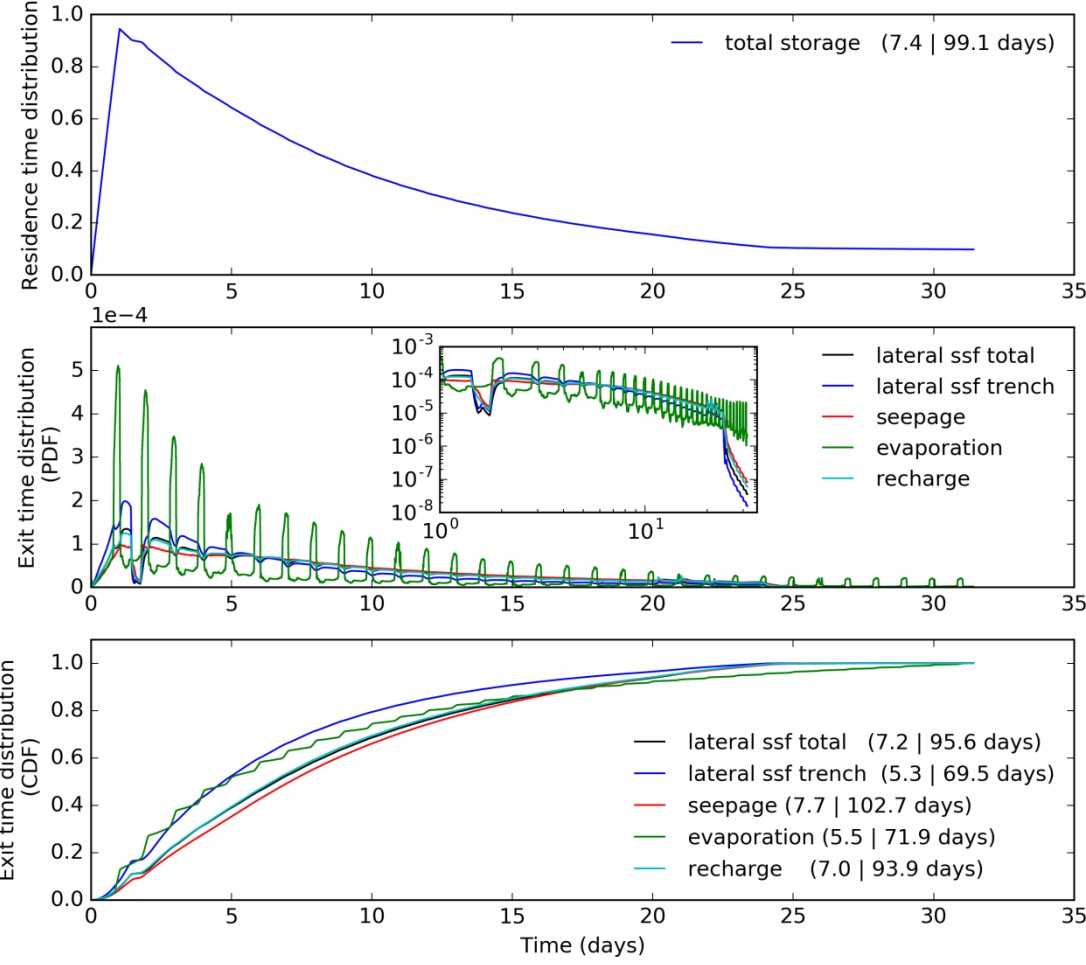

**Figure 7: Simulated residence and exit time distributions by Model 2. The values between parentheses refer to median residence or median travel times. The first value between parentheses refers to the actual days (of the field experiment), the second value refers to the amount of days based on the average daily rainfall amount for this site (natural conditions).**




**Figure 8: Relative concentration of δ²H in the unsaturated zone ($c_{un}(t)$) and the saturated zone ($c_{sat}(t)$), and relative subsurface flow (SSF) and recharge fluxes, at different time slices during the sprinkler experiment. Concentrations are relative to maximum concentration values at T = 20, and fluxes are relative to maximum flux values at T=370. Time (T) is h since the application of labeled sprinkler water.**




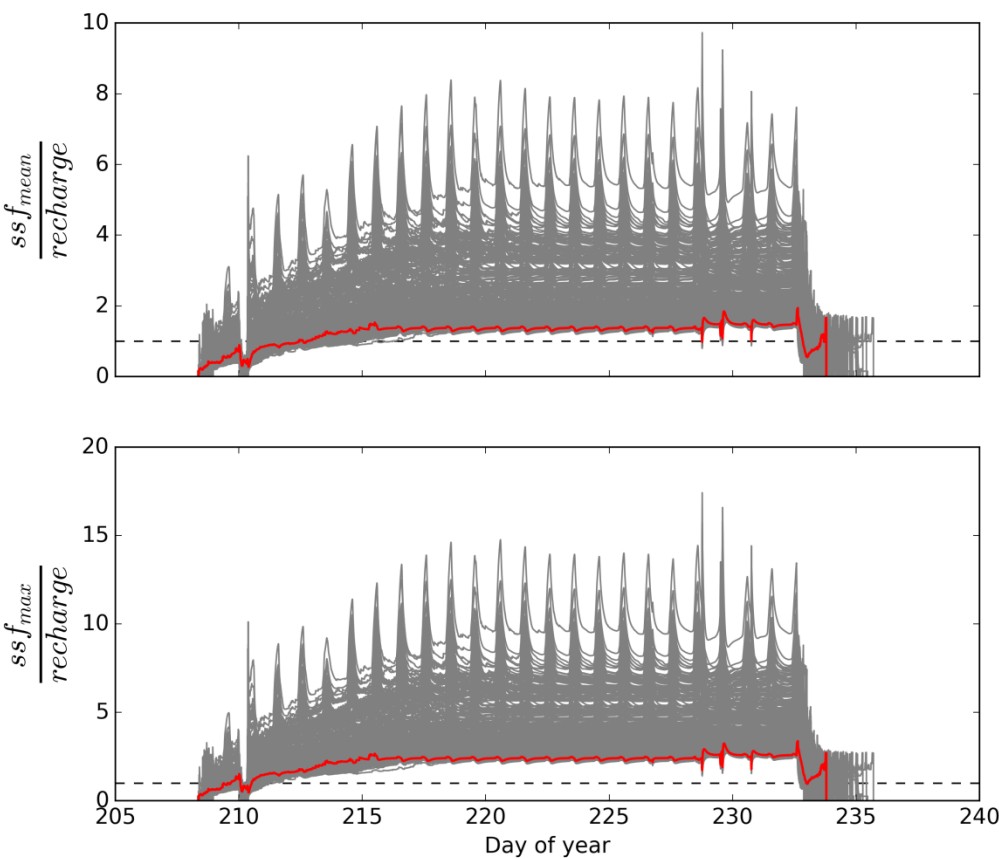

**Figure 9: Simulated ratios of (a) average subsurface flow and recharge fluxes, and (b) maximum subsurface flow and recharge fluxes, for the 308 behavioral parameter sets based on lateral subsurface flow and $\delta^2$H. The red line shows the result of Model 2 using the best parameter set in Table 3.**



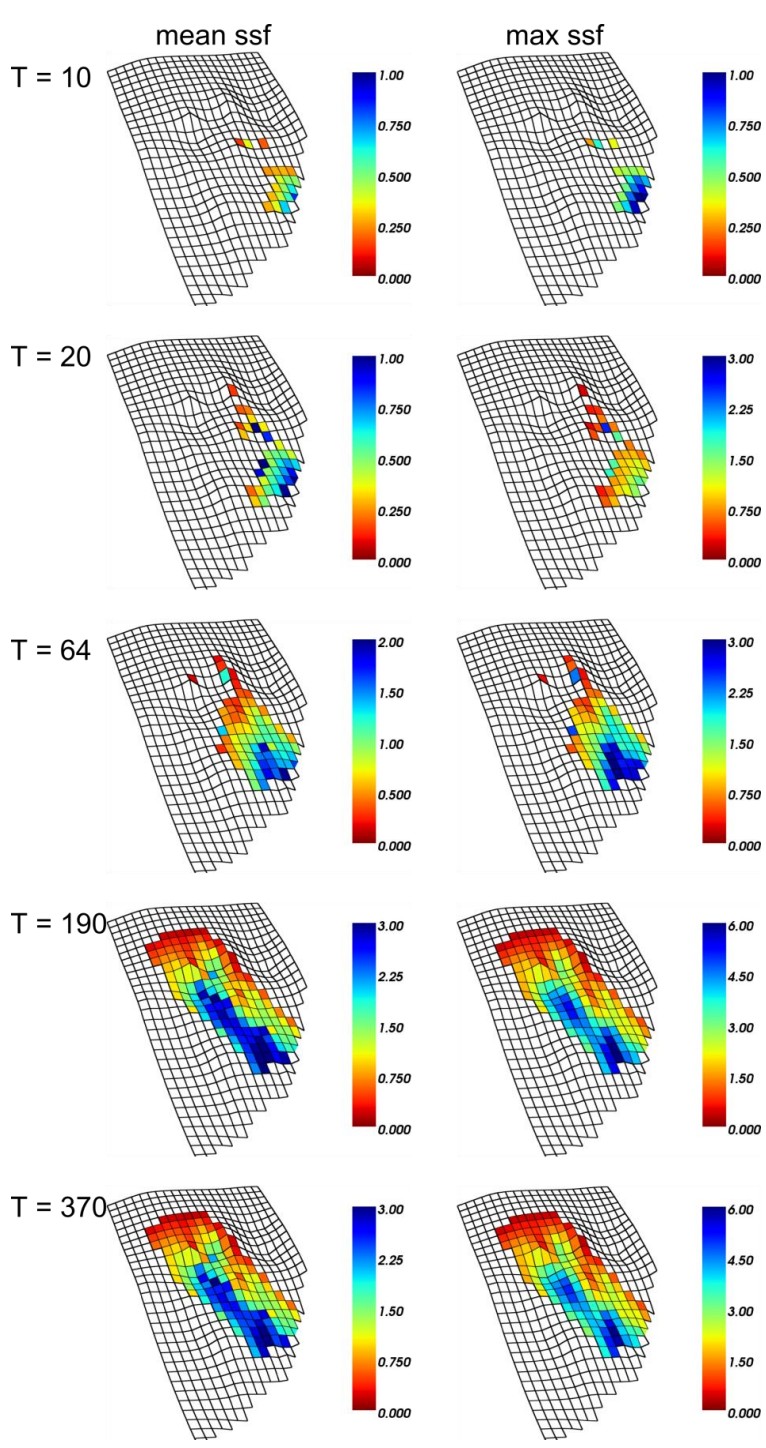

**Figure 10: Simulated ratios of average subsurface flow and recharge fluxes (ssf mean), and simulated ratios of maximum subsurface flow and recharge fluxes (ssf max) by Model 2 with the best parameter set. Time (T) is h since the application of labeled sprinkler water.**





**Table 4: Simulated maximum event contribution by Model 2 and the interval of maximum event contribution based on the 308 behavioral parameter sets.**

| Source of lateral subsurface flow | Maximum event contribution | Interval of maximum event contribution |
|---|---|---|
| Trench with deep groundwater mixing | 0.28 | 0.19 - 0.35 |
| Trench without deep groundwater mixing | 0.40 | 0.29 - 0.60 |
| Total lateral subsurface flow | 0.38 | 0.27 - 0.60 |





**Table 5: Tracer mass ($\delta^2$H) and water balance components as percentage of total input simulated by Model 2, and the minimum and maximum values for these balance components simulated by the 308 behavioral parameter sets based on lateral subsurface flow and $\delta^2$H. The time interval for these calculations was DOY 208 – DOY 243.**

|  | Unsaturated zone | Saturated zone | SSF (total) | SSF (trench) | Seepage | Evaporation |
|---|---|---|---|---|---|---|
| $\delta^2$H (%) | 9.6 | 0.0 | 52.0 | 31.1 | 26.1 | 12.4 |
| $\delta^2$H min - max (%) | 5.6 – 24.0 | 0.0 – 0.1 | 42.7 – 63.9 | 29.2 – 37.5 | 13.0 – 31.4 | 8.7 – 15.1 |
| Water (%) | 2.0 | 0.0 | 60.1 | 35.0 | 24.3 | 13.6 |
| Water min - max (%) | 0.0 – 14.4 | 0.0 – 0.2 | 54.3 – 65.2 | 33.8 – 37.3 | 12.6 - 28.3 | 10.7 - 14.5 |

