# Peer review of "A sprinkling experiment to quantify celerity-velocity differences at the hillslope scale"

_Hydrology and Earth System Sciences, 2017_

## Referee Comment (RC1) · Anonymous Referee #1 · 14 Apr 2017

The article reports an interesting sprinkling experiment on a natural hillslope. Isotopic signature and physical measurements are utilized to investigate the hillslope hydrograph, velocities, celerities, transit time distributions, and flowpaths. The site is well described and well studied as documented by an extensive literature review. In my opinion, the article is suitable for publication in Hess after minor revision.

In preparing a new version, the Authors may take into account the following suggestions:

1. Specify meteorological conditions during the sprinkling experiment. I believe no rainfall events were registered in the 24-days experiment.

2. The description of the different sensors location can be improved. I would also recommend to improve Figure 1 and introduce acronyms for each type of sensors in

the map.

3. In your findings, soil moisture responded earlier than lateral subsurface flow to irrigation. Please comment on this.

4. Section 4.3 is very relevant and I think it could be rewritten a bit more clearly. Could you also report results in a Figure/Table?

5. Table 2 could be improved by adding instrument depths.

6. Discussion on immobile soil water fraction should be expanded and stated more clearly.

7. Are estimated celerities compatible with WS10 response to storm events?

Finally, the manuscript present numerous typos and reference to figures and Tables is often incorrect. I suggest a thorough revision of the writing.

---

## Referee Comment (RC2) · Anonymous Referee #2 · 2 May 2017

This paper presents an experimenting and modelling study where a tracer is applied to a hillslope and its concentration is recorded in the outflow. I found the paper overall quite well written and structured, and the topic is interesting. Below I give a few suggestions for improvement:

1. Abstract. It is difficult to understand the significance of the study. The first sentence states that the difference between velocity and celerity is poorly understood. I would argue that it is very well understood. Perhaps the mechanisms explaining such differences are poorly understood. Many numbers are given in the abstract, but it is difficult to understand why such numbers would be interesting. I would stress in the abstract more the connection between experimenting and modelling, which I find the most interesting aspect of this work.

2. In the introduction, I would cite the paper "Velocity and celerity dynamics at plot scale inferred from artificial tracing experiments and time-lapse ERT", Journal of Hydrology, 2017, as it seems relevant for the study.

3. In section 3.1, before starting to describe what was done, it would be useful to illustrate why it was done it. What were the objectives of the experimental design? Which conditions did you want to recreate? Why? What about natural rainfall in addition to artificial rainfall? Did it happen? If not, what if it happened? Etc.

4. Equation 7 appears to be wrong – the integral of concentration is not equal to mass. What does Mout represent? If it is just the mass of tracer in the outflow, what about Evaporation? I guess the calculation of mean residence time should account also for this. . .

5. What was the recovery rate of water and tracers? Did it differ? Why?

6. Model 1 and Model 2 is a misleading terminology. In fact it appears that the model structure is the same, just the evaluation criteria are different.

7. The criteria used to calculate the behavioural parameter sets for Model 1 and 2 make the comparison difficult as the criteria with which the models are evaluated are very different. I think it would make more sense to make the behavioural parameters of Model 2 a subset of the behavioural parameters for Model 1, by requiring them to satisfy some additional constraints based on the tracers.

8. Indeed, in Figure 6 the two distributions do not appear to differ significantly, and in my opinion, given how they are constructed, they are not even comparable (besides the fact that the caption is not clear, in Figure 6 there are 2 green colors. Which green color do you mean?)

---

## Referee Comment (RC3) · Anonymous Referee #3 · 2 Jun 2017

I read the paper with interest. To me it is a well-structured and written case study giving a lot of experimental and modelling details. The experimental work is analysed based on data and a conceptual model and described in terms of celerity and velocity of water flow as response to a 24h sprinkling event. The experimental set up is impressive on the hillslope, but consists of only one isotopic (deuterium) tracer in the sprinkling water (constant for 24 hr). The topic fits very well in HESS and is, in my opinion, of interest to the audience of HESS.

I have only two suggestions and one minor point to take into consideration:

1) The discussion of the paper is entirely devoted to process understanding. However, I miss a section discussing the effects of the experimental set up (mainly the use of only one tracer in time and space). For example, what would the authors advice to improve

on the experimental set up and what would be the effect of applying multiple tracers (or spatially distributed or in time (think of also adding 18O, or others tracers like salts). It is not critics on the current work, but I think with so much emphasis on the experiment, it could be worthwhile to discuss that as well. This could maybe also be linked to the conclusion you draw that the "precise mechanism of disturbance transmittance remains unclear".

2) I would suggest the authors to rethink if parts of the paper cannot be transferred to appendix or supplement material. The paper is long and that distracts somewhat. Especially the field description (3.1-3.2) but even more the long model description including CRET and mixing model description, python etc info (3.7) and could be summarised in a few lines in the main article and all other details moved to the supplement material. To me that would be helpful.

3) There are quite some typo's and sloppiness, wrong references and inconsequent numbering of headings that should be rigorously checked by the authors before resubmitting

---

## Author Comment (AC1) · 30 Jun 2017

**Reply to Interactive comment by Anonymous Referee #1 "A sprinkling experiment to quantify celerity-velocity differences at the hillslope scale"**

Willem J. van Verseveld, Holly R. Barnard, Chris B. Graham, Jeffrey J. McDonnell, J. Renée Brooks, Markus Weiler

First of all we would like to thank Referee #1 for his/her evaluation of this manuscript and his/her suggestions will for sure improve this manuscript. Our answers (in blue) to the suggestions are written below each suggestion (black).

1. Specify meteorological conditions during the sprinkling experiment. I believe no rainfall events were registered in the 24-days experiment.

Yes, good point. We will describe this in more detail. Correct, no rainfall events occurred during the sprinkler experiment.

2. The description of the different sensors location can be improved. I would also recommend to improve Figure 1 and introduce acronyms for each type of sensors in the map.

We agree with the recommendation of the reviewer to improve Figure 1 and the associated description of the different sensors. We will modify the Figure and the associated text.

3. In your findings, soil moisture responded earlier than lateral subsurface flow to irrigation. Please comment on this.

Correct, soil moisture responded definitely earlier in the upper 30 cm. Below 60 cm soil moisture responded later than lateral subsurface flow. Apparently, antecedent wetness conditions were such (in combination with the soil depth profile increasing upslope, and applied irrigation intensity) that we observed this response pattern. At the same hillslope, McGuire and McDonnell (2010) found during wet conditions (winter) that peak hillslope runoff always lagged peak soil moisture at about 5 h.

4. Section 4.3 is very relevant and I think it could be rewritten a bit more clearly. Could you also report results in a Figure/Tabley

Yes, we agree it would be helpful to include the results in a Table of this paragraph, and rewrite this section a bit more clearly.

5. Table 2 could be improved by adding instrument depths.

Since this table is providing information on average vertical velocities (sampling depth/ response time), and already contains quite some number of columns we prefer to leave the instrument depths out of Table 2.

6. Discussion on immobile soil water fraction should be expanded and stated more clearly.

We agree with the reviewer that this is an important part of the Discussion, and we will elaborate more on the immobile soil water fraction and the link to our modeling and experimental results in the Discussion.

7. Are estimated celerities compatible with WS10 response to storm events?

Quite an interesting question but at the same time difficult to answer. Please note that our reported celerities are vertical celerities through the soil profile, and not lateral celerities, a component one also needs to take into account for lateral subsurface flow or streamflow. Furthermore the response time (or celerity) is also a function of antecedent wetness conditons, for example McGuire and McDonnell (2010) found during the winter period response times of 0.3-0.5 h for water content reflectometers at 100 cm depth, and thus much higher celerities than observed during our sprinkler experiment. Additionally, McGuire and McDonnell (2010) did find that hillslope peak runoff lagged soil moisture responses at 100 cm depth and different positions (up till > 25 m upslope) by about 5 h, indicating very fast lateral celerities (much higher than their observed vertical celerities). Although outside of the scope of our presented work, it would definitely be interesting to investigate this in more detail for the hillslope and WS10, for example by comparing the timing of peak rainfall or mass center of rainfall and the hillslope lateral subsurface flow and WS10 streamflow response.

Finally, the manuscript present numerous typos and reference to figures and Tables is often incorrect. I suggest a thorough revision of the writing.

Yes, we will make sure to correct typos and incorrect references to Figures and Tables.

---

## Author Response (AR1)

**Reply to Interactive comment by Anonymous Referee #1 "A sprinkling experiment to quantify celerity-velocity differences at the hillslope scale"**

Willem J. van Verseveld, Holly R. Barnard, Chris B. Graham, Jeffrey J. McDonnell, J. Renée Brooks, Markus Weiler

First of all we would like to thank Referee #1 for his/her evaluation of this manuscript and his/her suggestions will for sure improve this manuscript. Our answers (in blue) to the suggestions are written below each suggestion (black).

1. Specify meteorological conditions during the sprinkling experiment. I believe no rainfall events were registered in the 24-days experiment.

Yes, good point. We will described this in more detail on page 5, line numbers 3-6.

2. The description of the different sensors location can be improved. I would also recommend to improve Figure 1 and introduce acronyms for each type of sensors in the map.

We agree with the recommendation of the reviewer to improve Figure 1 and the associated description of the different sensors. We did modify the Figure and the associated text on page 5.

3. In your findings, soil moisture responded earlier than lateral subsurface flow to irrigation. Please comment on this.

Correct, soil moisture responded definitely earlier in the upper 30 cm. Below 60 cm soil moisture responded later than lateral subsurface flow. Apparently, antecedent wetness conditions were such (in combination with the soil depth profile increasing upslope, and applied irrigation intensity) that we observed this response pattern. At the same hillslope, McGuire and McDonnell (2010) found during wet conditions (winter) that peak hillslope runoff always lagged peak soil moisture at about 5 h.

4. Section 4.3 is very relevant and I think it could be rewritten a bit more clearly. Could you also report results in a Figure/Tabley

Yes, we agree and did report the results in Figure 5. We also rewrote this section a bit more clearly.

5. Table 2 could be improved by adding instrument depths.

Since this table is providing information on average vertical velocities (sampling depth/ response time), and already contains quite some number of columns we prefer to leave the instrument depths out of Table 2.

6. Discussion on immobile soil water fraction should be expanded and stated more clearly.

We agree with the reviewer that this is an important part of the Discussion. We did expand the Discussion on page 15, line numbers 10-15.

7. Are estimated celerities compatible with WS10 response to storm events?

Quite an interesting question but at the same time difficult to answer. Please note that our reported celerities are vertical celerities through the soil profile, and not lateral celerities, a component one also needs to take into account for lateral subsurface flow or streamflow. Furthermore the response time (or celerity) is also a function of antecedent wetness conditons, for example McGuire and McDonnell (2010) found during the winter period response times of 0.3-0.5 h for water content reflectometers at 100 cm depth, and thus much higher celerities than observed during our sprinkler experiment. Additionally, McGuire and McDonnell (2010) did find that hillslope peak runoff lagged soil moisture responses at 100 cm depth and different positions (up till > 25 m upslope) by about 5 h, indicating very fast lateral celerities (much higher than their observed vertical celerities). Although outside of the scope of our presented work, it would definitely be interesting to investigate this in more detail for the hillslope and WS10, for example by comparing the timing of peak rainfall or mass center of rainfall and the hillslope lateral subsurface flow and WS10 streamflow response.

Finally, the manuscript present numerous typos and reference to figures and Tables is often incorrect. I suggest a thorough revision of the writing.

Yes, we did make sure to correct typos and incorrect references to Figures and Tables.

**Reply to Interactive comment by Anonymous Referee #2 "A sprinkling experiment to quantify celerity-velocity differences at the hillslope scale"**

Willem J. van Verseveld, Holly R. Barnard, Chris B. Graham, Jeffrey J. McDonnell, J. Renée Brooks, Markus Weiler

First of all we would like to thank Referee #2 for his/her time, useful comments on this manuscript and asking relevant questions. His/her suggestions will for sure help to improve the quality of this manuscript. Our answers (in blue) to the suggestions are written below each suggestion (black).

1. Abstract. It is difficult to understand the significance of the study. The first sentence states that the difference between velocity and celerity is poorly understood. I would argue that it is very well understood. Perhaps the mechanisms explaining such differences are poorly understood. Many numbers are given in the abstract, but it is difficult to understand why such numbers would be interesting. I would stress in the abstract more the connection between experimenting and modelling, which I find the most interesting aspect of this work.

We fully agree with the reviewer that the significance of the study is not very clear from the abstract alone, and that indeed the connection between experimenting and modelling is an interesting aspect of our study. We did change the abstract to make the significance of our study clearer, also leaving out the many numbers. Additionally, we did also change the first sentence.

2. In the introduction, I would cite the paper "Velocity and celerity dynamics at plot scale inferred from artificial tracing experiments and time-lapse ERT", Journal of Hydrology, 2017, as it seems relevant for the study.

This paper seems indeed relevant for our study, and we did include it in the Introduction, page 3, line numbers 14-16.

3. In section 3.1, before starting to describe what was done, it would be useful to illustrate why it was done it. What were the objectives of the experimental design? Which conditions did you want to recreate? Why? What about natural rainfall in addition to artificial rainfall? Did it happen? If not, what if it happened? Etc.

We think the objectives of this study are clearly mentioned in section 1, also linked to the experimental design. We agree that we did not include information on natural rainfall during the sprinkler experiment (also mentioned by another reviewer), and on what kind of conditions we did want to recreate (in comparison with natural conditions). We did add this information, see page 5, line numbers 3-6.

4. Equation 7 appears to be wrong – the integral of concentration is not equal to mass. What does Mout represent? If it is just the mass of tracer in the outflow, what about Evaporation? I guess the calculation of mean residence time should account also for this. . .

We agree that Equation 7 is not clearly explained, but it is correct, under steady-state water flow conditions (the exit distribution can then be calculated from concentrations alone). Mout

is the mass of tracer at the exit boundaries (for evapotranspiration, recharge, seepage and lateral subsurface flow (captured at trench and total). We did calculate the exit time distribution for these exit boundaries.

Calculation of the mean residence time distribution was based on mass in the storage zone (unsaturated and saturated zone), Equation 8. Outgoing mass fluxes from evaporation and later subsurface flow are thus taken into account.

We did improve the description how we did calculate the exit time distributions (page 10, line number 25) and Equation 7 was changed; only the right part of the equation (mass) is used now, because of unsteady water flow conditions during our sprinkler experiment.

5. What was the recovery rate of water and tracers? Did it differ? Why?

The recovery rates of (simulated) water and tracers are presented in Table 5 and did differ, both at the trench and for total later subsurface flow, with higher recovery rates for water. We explain this difference in the paper by a dual porosity system; about 10% of deuterium remains in the unsaturated zone.

6. Model 1 and Model 2 is a misleading terminology. In fact it appears that the model structure is the same, just the evaluation criteria are different.

Yes, the model structure is the same. We think this depends on the definition, are two models with two different parameter sets, but the same model structure, different models? We think the models are different, and do not think Model 1 and 2 are misleading terms, as we need to make clear in the text to which model (or model with a specific evaluation criteria) we are referring to.

7. The criteria used to calculate the behavioural parameter sets for Model 1 and 2 make the comparison difficult as the criteria with which the models are evaluated are very different. I think it would make more sense to make the behavioural parameters of Model 2 a subset of the behavioural parameters for Model 1, by requiring them to satisfy some additional constraints based on the tracers.

We don't think it makes more sense to constrain the parameter set of Model 2 by the behavioural parameters of Model 1. By doing so, one could a-priori exclude possible behavioural parameter sets for Model 2. Please also note that Model 1 is still part of the behavioral parameter sets of Model 2 (we did not reject Model 1 based on based on the objective criterion for the deuterium breakthrough).

8. Indeed, in Figure 6 the two distributions do not appear to differ significantly, and in my opinion, given how they are constructed, they are not even comparable (besides the fact that the caption is not clear, in Figure 6 there are 2 green colors. Which green color do you mean?)

Correct, the two distributions for most parameters of Model 1 and Model 2 are similar. However, parameters n, n0 and kb are more identifiable for Model 2 (from Figure 6 and lower parameter uncertainty (Table 3)). Please note that we use two different colors in Figure 6: green and blue. Green is transparent, so where the color is dark green, the two distributions overlap. We did make this clearer in the legend of Figure 6.

**Reply to Interactive comment by Anonymous Referee #3 "A sprinkling experiment to quantify celerity-velocity differences at the hillslope scale"**

Willem J. van Verseveld, Holly R. Barnard, Chris B. Graham, Jeffrey J. McDonnell, J. Renée Brooks, Markus Weiler

First of all we would like to thank Referee #3 for his/her time and useful comments on this manuscript. His/her suggestions will for sure help to improve the quality of this manuscript. Our answers (in blue) to the suggestions are written below each suggestion (black).

1) The discussion of the paper is entirely devoted to process understanding. However, I miss a section discussing the effects of the experimental set up (mainly the use of only one tracer in time and space). For example, what would the authors advice to improve on the experimental set up and what would be the effect of applying multiple tracers (or spatially distributed or in time (think of also adding 18O, or others tracers like salts). It is not critics on the current work, but I think with so much emphasis on the experiment, it could be worthwhile to discuss that as well. This could maybe also be linked to the conclusion you draw that the "precise mechanism of disturbance transmittance remains unclear".

Yes, we like the idea of adding a paragraph to the paper on possible future work on this topic, including the experimental set up (e.g. use of multiple tracers, applying a disturbance through higher rainfall intensities, lateral celerities etc.). We added paragraph 5.5 (page 19) to the paper, it is about the connection between field experiments and modeling, and possible other experimental setups and future work at this site.

2) I would suggest the authors to rethink if parts of the paper cannot be transferred to appendix or supplement material. The paper is long and that distracts somewhat. Especially the field description (3.1-3.2) but even more the long model description including CRET and mixing model description, python etc info (3.7) and could be summarized in a few lines in the main article and all other details moved to the supplement material. To me that would be helpful.

We agree that the paper is long and transferring parts of the paper you mention to appendix or supplement material could be helpful to readers. We are not sure if transferring parts of the Methods to supplement material is common for HESS. In the end we did decide not to transfer parts of the Methods section (the only section we think makes sense to transfer to appendix or supplement material) of the paper to appendix or supplement material. While for some readers a long Methods section may be distracting, for other readers going back and forth between the paper and appendix or supplement material may be somewhat distracting.

3) There are quite some typo's and sloppiness, wrong references and inconsequent numbering of headings that should be rigorously checked by the authors before resubmitting

Yes, we did make sure to correct typos and incorrect references to Figures and Tables.

[revised manuscript text omitted]
 (%) | 0.0 – 14.4 | 0.0 – 0.2 | 54.3 – 65.2 | 33.8 – 37.3 | 12.6 - 28.3 | 10.7 - 14.5 |